# ON EXPLAINING NEURAL NETWORK ROBUSTNESS WITH ACTIVATION PATH

**Ziping Jiang** *
School of Computing and Communications, Lancaster University
{z.jiang7}@lancaster.ac.uk

## ABSTRACT

Despite their verified performance, neural networks are prone to be misled by maliciously designed adversarial examples. This work investigates the robustness of neural networks from the activation pattern perspective. We find that despite the complex structure of the deep neural network, most of the neurons provide locally stable contributions to the output, while the minority, which we refer to as float neurons, can greatly affect the prediction. We decompose the computational graph of the neural network into the fixed paths and float paths and investigate their role in generating adversarial examples. Based on our analysis, we categorize the vulnerable examples into Lipschitz vulnerability and float neuron vulnerability. We show that the boost of robust accuracy from randomized smoothing is the result of correcting the latter. We then propose an SC-RFP (smoothed classifier with repressed float path) to further reduce the instability of the float neurons and show that our result can provide a higher certified radius as well as accuracy.

## 1 INTRODUCTION

Despite their verified performance, neural networks are prone to be misled by maliciously designed adversarial examples. In response to this issue, many studies focus on defensive algorithms that aim to increase the robustness of deep neural networks. One of the emerging topics in this field is certifiable methods that aim to construct a guaranteed region, within which classifiers are able to provide stable results regardless of the perturbation. The certifiable methods appear in two different forms: verifiable training and randomized smoothing.

This work introduces an SC-RFP (smoothed classifier with repressed float path) which builds on randomized smoothing algorithms and is able to further improve their robustness accuracy. We decompose the local mapping function into fixed paths and float paths according to the stability of neurons on the path. The fixed paths have a stable mapping relationship between input and output, while the float paths can result in a sudden change of the mapping function and alter the result. We categorize the adversarial examples into Lipschitz vulnerable and float neuron vulnerable. With respect to the ability of randomized classifiers in correcting misclassified data, we conclude that the essence of the smoothed classifier is to average the contribution of the float path and achieve a locally stable result. Based on this, we further repress the float paths of the network and show that such a classifier can achieve better performance.

The theoretical basis of this work is developed from the analysis of the activation region that was initially proposed for explaining the performance of neural network with a piecewise linear activation function. The input domain of such a neural network $N$ is separated into many regions, within which the mapping of $N$ is piecewise linear. Previous investigation of this field includes the expressivity, sensitivity, and potential issues of the network. However, due to the complexity of neural network, the theoretical investigation only provides insights into the neural network but has yet to be deployed downstream. In this work, we use the theory to explain the model robustness and introduce a novel way to apply the complex theory to practical.

The contributions of this work are: (1) we introduce a complete framework to describe and decompose the neural network according to the activation status of each neuron; (2) we provide an explana-

---

*Ziping Jiang is with the LIRA Center, Lancaster University.

tion of adversarial examples and discuss the role of smoothed classifiers as well as their contribution in correcting misclassified example; (3) we introduce SC-RFP that achieves better performance in certifying the network.

## 2 RELATED WORKS

The adversarial examples are malicious inputs that are formed by applying an imperceptible perturbation to the original inputs but result in misclassification of a well-trained network (Biggio et al. (2013); Szegedy et al. (2013)). To explain the existence of adversarial example, previous works presented several hypotheses, such as *linearity hypothesis* (Szegedy et al. (2013); Luo et al. (2015)) and *Evolutionary stalling* (Rozsa et al. (2016)). Early works on increasing the robustness of neuron networks focused on adversarial training methods (Goodfellow et al. (2014); Wong et al. (2020); Tramèr et al. (2017); Dong et al. (2018); Kurakin et al. (2016)), while recent investigation shows adversarial training methods can be broken by more advanced attacks.

To address the issue, certifiable training and randomized smoothing methods aim to provide a certified region, within which the input data are free from attack. By viewing the training as a convex optimization problem, dual relaxation approaches apply duality to provide a solid bound for training as well as verify the network (Wong & Kolter (2018); Wong et al. (2018)). An alternative is to estimate the Lipschitz boundary of the network and introduce constraints on either objective loss (Tsuzuku et al. (2018)) or forward propagation (Lee et al. (2020); Weng et al. (2018); Zhang et al. (2019); Huang et al. (2021)). As verifiable training methods often come with a compromise of performance, recent works focus on bridging the gap between adversarial and verifiable training to address the scalability and accuracy issue (Xiao et al. (2018); Balunović & Vechev (2020); De Palma et al. (2022)).

On the other hand, randomized smoothing introduces a smoothed classifier to the base classifier, therefore has a limited effect on the performance of standard models. Cao & Gong (2017) first propose to ensemble the information around input data to smooth the prediction, but fail to provide a theoretical guarantee on the result. Lecuyer et al. (2019) certify the result of the smoothed classifier with differential privacy. Cohen et al. (2019) provides a theoretical analysis of the certifiable with Monte Carlo, followed by Levine et al. (2019); Li et al. (2019). Jeong & Shin (2020) introduces a regularized to improve the prediction constantly over noise, Jeong et al. (2021) trains the model on a convex combination of samples and Salman et al. (2019) employs PGD attack with randomized smoothing to further increases the robustness accuracy.

Another related topic is the explainability of neuron networks. Lin et al. (2017), Hornik et al. (1989) and Park et al. (2020) investigate how deep models approximate an objective function. An inspiring observation is that the network is a piecewise function when the activation function is piecewise linear (Pascanu et al. (2013)). The number of linear regions is then adopted as a proxy of network complexity (Montufar et al. (2014); Hanin & Rolnick (2019a;b)). Novak et al. (2018) studies the network sensitivity by countering the transition density of trajectory in the input space. Jiang et al. (2022) compares the similarity of activation patterns globally to study the limitation of deep neural network. Inspired by the theoretical investigation, Jordan et al. (2019) introduces an algorithm named GeoCert that computes the $l_p$ bound of the network with a piecewise linear activation function. Zhang et al. (2022) proposes an algorithm that systematically searches the adversarial example based on the activation space of ReLU network.

## 3 PRELIMINARIES

### 3.1 NOTATIONS

Let $\mathcal{N}$ be a $d$ block feedforward neural network for classification task with measure zero parameter set $\theta$ with respect to Lebesgue measure. Each of the block $h_i$ consists of a linear affine $\phi_i$, an optional batch-normalization layer $\psi_i$, and piecewise linear activation function $\sigma_i$, while the last block $h_d$ omits the activation function. Consider $D$ as the distribution of a classification problem with $c$ classes from $R^{n_0}$ to $Y = \{1, 2, \ldots, c\}$, network $N$ computes a function $f : R^{n_0} \to R^c$, where $f$ is a composition of $d$ blocks $f = h_n \circ h_{d-1} \cdots \circ h_1$.

For every $(x, y) \sim D$, the network computes a probability for each class $f(x) \in R^c$ and predicts the label of $x$ as the class with the highest probability: $\hat{y} = \arg\max_{m \in Y} f_m(x)$, where $f_m(x)$ is the $m$-th element of the network output vector. We use $x_i(x; \theta)$, $y_i(x; \theta)$ and $z_i(x; \theta)$ to denote the input, output, and pre-activation value of block $i$ for data $x$. Tuple pair $(i, j)$ and set $\mathcal{I} = \cup_{i=1}^{d-1}\{(i, j)|j \in \{1, \dots, n_i\}\}$ denote the $j$-th neuron of layer $i$ and all the neurons in intermediate layers respectively, where $n_i$ is the output size of layer $i$.

## 3.2 RANDOMIZED CERTIFIABLE CLASSIFIER

The research on *certifiable training* aims to provide a guaranteed region for its input $x$, within which a classifier always provide stationary result. To be specific, a classifier is regarded as robust for an input $x$ for perturbation of size $r$ if:

$$\arg\max_{m \in Y} f_m(x') = \arg\max_m f_m(x), \forall x' \in B_p(x, r) \tag{1}$$

where $B_p(x, r) := \{x' : \|x' - x\|_p \le r\}$ is the sphere with radius $r$ measured by the metric induced by $p$-norm. By taking the standard performance of the classifier into account, the robust accuracy of $f$ with radius $r$ is then defined as:

$$R(f) = \mathbb{E}_{(x,y) \sim D}\left[\arg\max_{m \in Y} f_m(x') = y, \forall x' \in B_p(x, r)\right]. \tag{2}$$

However, robust models often incur increased stableness and impaired expressivity. As a concession for that, randomized algorithms are proposed to verify the network with a sound theoretical bound at the cost of slight additional computation other than model performance.

Let $g$ be a randomized algorithm constructed based on classifier $f$. Given a data $(x, y) \sim D$, $g$ employs a certain degree of randomness during the induction of $f$. For instance, smoothed classifier $g$ computes the probability of $f(x + \epsilon)$ belongs to class $i$ given $\epsilon \in \mathcal{N}(0, \sigma^2 I)$(Cohen et al. (2019)):

$$g_i(x) = \mathbb{P}(\arg\max_{m \in \mathcal{Y}} f_m(x + \epsilon) = i), \epsilon \sim \mathcal{N}(0, \sigma^2 I), \tag{3}$$

With certain confidence level $\alpha$, the lower bound of $\underline{p_A}$ of random variable $g_y(x)$ and the upper bound $\overline{p_B}$ of the probability of second possible class $\max m \in Y g_y(x)$ can be computed. This induces a certified radius of classifier $g(x)$:

$$r = \frac{\sigma}{2}\left(\Phi^{-1}(\underline{p_A}) - \Phi^{-1}(\overline{p_B})\right). \tag{4}$$

For every $x' \in B_2(x, r)$, if $\arg\max_{m \in Y} g_m(x) = Y$, then $\arg\max_{m \in Y} g_m(x') = y$. In other words, with confidence level $\alpha$, every $x'$ within the radius can be correctly classifier by the smoothed classifier. The smoothed classifier can also appear in other forms depends on the randomness and training algorithm (Levine et al. (2019); Salman et al. (2019); Jeong & Shin (2020)).

## 3.3 ACTIVATION PATTERN AND COMPUTATIONAL PATH

The input space is partitioned into small linear regions by a neural network with piecewise linear activation function. Each of the regions is initially referred to as an activation region. Previous works mostly focused on the general properties of linear regions in investigating the expressivity and limitation of neural network. As our objective is to study the unit-wise reaction towards perturbation, introducing additional notations to describe the neurons is necessary for our analysis.

**Definition 1** (Generalized Activation Pattern / Region). *Let $\mathcal{N}$ be a network defined as Section 3.1. Denote $\Gamma = \{\gamma_1, \gamma_2, \dots, \gamma_q\}$ as a set of breakpoints that separates the domain of activation function into $q + 1$ intervals $U = \{U_0, U_1, \dots, U_q\}$. A generalized activation pattern of $\mathcal{N}$ is an assignment to each neuron of a label:*

$$\mathcal{A} := \{a_{ij} \,|a_{ij} \in \{0, 1, \dots, q\}, (i, j) \in \mathcal{I}\}.$$

*Given an activation pattern $\mathcal{A}$, the activation region of $\mathcal{A}$ is defined as:*

$$\mathcal{R}(\mathcal{A}; \theta, \sigma, \Gamma) := \{x \in R^{in}|z_{ij}(x; \theta) \in U_{a_{ij}}, a_{ij} \in \mathcal{A}\}.^1 \tag{5}$$

**Activation Region Operator.** The generalized activation pattern describes the activation status of each unit at the intermediate layers. Given an activation pattern $\mathcal{A}$ and fixed dependencies, $\mathcal{R}(\mathcal{A})$ can be viewed as an operator that finds the region, such that, for every $x \in \mathcal{R}(\mathcal{A})$, the pre-activation value of each neuron $z_{ij}(x; \theta)$ locates within the interval $U_{a_{ij}}$ that determined by its pattern $a_{ij} \in \{0, 1, \dots, q\}$. However, as the scale of the network grows, the mean volume of a single activation region decreases exponentially and can hardly provide insights into model robustness. Therefore, it is necessary to generalize our investigation from single region to its neighbor.

**From Single Region to its Neighbor.**[1] Equation 5 suggests that the essence of operator $\mathcal{R}(\cdot)$ is to find all the $x$ that satisfies a certain constraint determined by the activation pattern. It can be expressed as the intersection of a sequence of subspaces:

$$\mathcal{R}(\mathcal{A}) = \bigcap_{\forall i,j} \{x \in R^{in} | z_{ij}(x) \in U_{a_{ij}}\}. \tag{6}$$

Given a subset of indexes, removing the intersection operations on those neurons defines a larger region that contains $\mathcal{R}(\mathcal{A})$. This implies that adjacent activation regions can be merged into one by releasing constraints on certain neurons. We described it with an incomplete activation pattern:

**Definition 2.** *Let $\mathcal{N}$ be a network defined as Section 3.1. Given an activation pattern $A$ and a subset of the index set $\mathcal{I}^c \in \mathcal{I}$, we denote $\mathcal{A}_{\mathcal{I}^c} \subset \mathcal{A}$ is an **incomplete activation pattern** of $A$:*

$$\mathcal{A}_{\mathcal{I}^c} := \{a_{ij} \, | a_{ij} \in \mathcal{A}, (i,j) \in \mathcal{I}^C\}.$$

*The merged activation region of $\mathcal{A}_{\mathcal{I}^c}$ is denoted as:*

$$\mathcal{R}(\mathcal{A}_{\mathcal{I}^c}) = \bigcap_{(i,j) \in \mathcal{I}^c} \{x \in R^{in} | z_{ij}(x) \in U_{a_{ij}}\}.$$

In the following section, we delve into such merged region to investigate model robustness.

## 4 FROM FLOAT PATH TO MODEL ROBUSTNESS

### 4.1 FLOAT NEURON AND PATH

Merged activation regions are irregularly shaped as it is defined by the post-activation of neurons, while a regular subspace, such as sphere $B(x, r)$, is preferred in analyzing the properties of a network. Definition 3 introduces float neurons and fixed neurons to describe the status of neurons in a regular subspace, followed by Lemma 1 which builds a connection between Definitions 2 and 3.

**Definition 3** (Float and Fixed Neuron). *Let $\mathcal{N}$ be a network defined as Section 3.1. For any space $R \subset R^{n_0}$, if a neuron $z$ has the same pattern for any $x \in R$, we refer it as a fixed neuron of $R$, otherwise it is a float neuron. We denote the collection of fixed neurons and float neurons in region $R$ as $\mathcal{I}^X(R)$ and $\mathcal{I}^T(R)$, respectively:*

$$\mathcal{I}^X(R) = \{(i,j) \mid \exists x_1, x_2 \in R, a_{ij}(x_1) \neq a_{ij}(x_2)\}$$
$$\mathcal{I}^T(R) = \{(i,j) \mid \forall x_1, x_2 \in R, a_{ij}(x_1) = a_{ij}(x_2)\}.$$

The following lemma shows that: (1) the fixed and float neurons are complementary sets in $\mathcal{I}$ and (2) any subset $R \in R^{n_0}$ can be covered by a merged region defined by the fixed neurons of $R$.

**Lemma 1.** *Let $\mathcal{N}$ be a neural network defined as Section 3.1. Given $R \subset R^{n_0}$, denote $\mathcal{I}^X$ and $\mathcal{I}^T$ are the set of float neuron and fixed neuron in $R$. Then:*

*1. $\mathcal{I}^X(R) \bigcup \mathcal{I}^T(R) = \mathcal{I}$,*  *2. $R \subset \mathcal{R}(\mathcal{A}_{\mathcal{I}^X})$.*

Now that we have described the status of neurons and illustrated their geometric relationship, we introduce following definitions with the goal of decomposing the computational graph of $\mathcal{N}$.[2]

---

[1]Appendix A provides additional discussion regarding activation pattern and geometric intuition.
[2]Appendix B explains the computational graph of network with more details.

**Definition 4** (Path). *Let $\mathcal{N}$ be a network defined as Section 3.1. A path of $\mathcal{N}$ is a set of neurons:*

$$\zeta := \{(i, \zeta_i) | i = 0, 1, \ldots, d\}, \zeta_i \in \{1, 2, ..., n_i\}, \tag{7}$$

*where $n_i$ is the number of neurons in layer $i$. The value of a path is defined as:*

$$\zeta(x, \mathcal{A}) := x_{\zeta_0} \prod_{m=1}^{i} d_{a_{m,\varsigma_m}} W'^{(m)}_{(\zeta_m, \zeta_{m-1})}, x \in R, \tag{8}$$

*where $x_{\zeta_j}$ is the $\zeta_j$-th element of input, $W'^{(i)}$ is the equivalent matrix linear transformation $\psi_i \circ \phi_i$, $a_{m,\varsigma_m}$ is the activation pattern of neuron $(m, \zeta_m)$ and $d_{a_{m,\varsigma_m}}$ is the slope of activation $\sigma$ within $a_{m,\varsigma_m}$-th interval.*

**Definition 5** (Float Path and Fixed Path). *Let $\mathcal{N}$ be a neural network defined as Section 3.1. Given a subspace $R \in R^{n_0}$, a path $\zeta$ of $\mathcal{N}$ is a float path in $R$ if there exist a neuron $(i, \zeta_i) \in \zeta$ is a float neuron, otherwise it is a fixed path:*

$$\begin{aligned} \text{float path in } R &:= \{\zeta | \exists \zeta_i \in \zeta, (i, \zeta_i) \notin \mathcal{I}^I(R)\} \\ \text{fixed path in } R &:= \{\zeta | \forall \zeta_i \in \zeta, (i, \zeta_i) \in \mathcal{I}^I(R)\}. \end{aligned} \tag{9}$$

*The value of float paths and float paths in $R$ are denoted as:*

$$\mathcal{Z}^T(x, \mathcal{A}; R) = \sum_{\text{float path in } R} \zeta(x, \mathcal{A}), \qquad \mathcal{Z}^T(x, \mathcal{A}; R) = \sum_{\text{float path in } R} \zeta(x, \mathcal{A}).$$

Figure 1 illustrate the proposed concepts with a simple network with 2D input, 4D output and 1 hidden layer. Neuron $(1, 4)$ is the only float neuron in region $B(x, r)$. As all the other neurons are fixed, the only non-linearity is provided by neuron $(1, 4)$. The computational graph of $\mathcal{N}$ can be decomposed into *float paths* and *fixed paths* according to whether $(1, 4)$ is on the path.

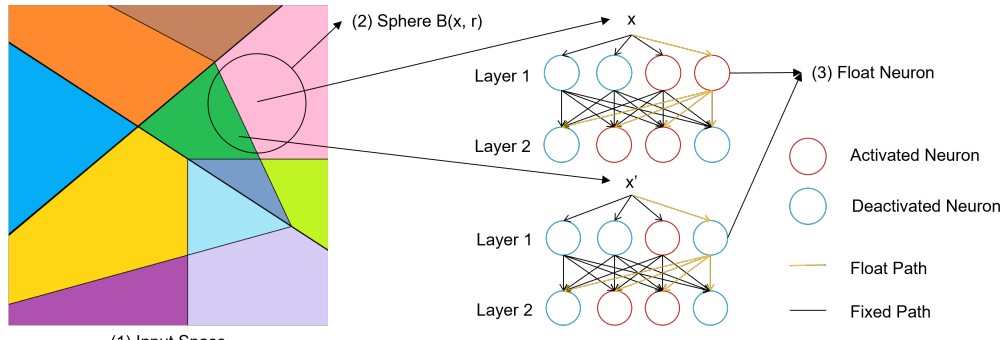

Figure 1: An illustration of the fixed (float) path and neuron of a neural network with 2D input, 4D output, and 1 hidden layer. (1) The 2D input space; (2) A sphere centered at $x$; (3) A float neuron with index (1,4) in the network.

By removing the constraint of neuron $(1, 4)$ in Equation 6, an incomplete activation pattern $\mathcal{A}_{\mathcal{I}^c}$ defines a merged activation region $\mathcal{R}(\mathcal{A}_{\mathcal{I}^x})$ from green and pink, where $\mathcal{I}^c = \mathcal{I}/\{(1, 4)\}$ is subset of neurons without neuron $(1, 4)$. As Lemma 1 suggests, region $\mathcal{R}(\mathcal{A}_{\mathcal{I}^x})$ covers $B(x, r)$. Moreover, according to Definition 5, the activation pattern of neurons on fixed paths remains unchanged for every $x \in \mathcal{R}(\mathcal{A}_{\mathcal{I}^x})$. Therefore, given $x, x' \in B(x, r)$ with different activation patterns, the value of their fixed paths is linear according to Equation 8. The following theorem generalizes the above discussion by aggregating the value of all fixed and float paths with Definition 5.

**Theorem 1.** *Let $\mathcal{N}$ be a neural network defined as Section 3. Given $R \subset R^{n_0}$, the following statements hold for any $x, x' \in R$ with activation pattern $\mathcal{A}$ and $\mathcal{A}'$:*

*1. $f(x) = \mathcal{Z}^I(x, \mathcal{A}; R) + \mathcal{Z}^T(x, \mathcal{A}; R)$*

2. $f(x) - f(x') = J(x)(x - x') + \mathcal{Z}^T(x', \mathcal{A}; R) - \mathcal{Z}^T(x', \mathcal{A}'; R)$

*where $J(x)$ is the Jacobian matrix of $f$ at $x$.*

Statement 1 decomposes the computational graph of $f(x)$ into a linear function $\mathcal{Z}^I(x, \mathcal{A}; \mathcal{R})$ and an unstable function $\mathcal{Z}^T(x, \mathcal{A}; \mathcal{R})$ with high non-linearity. Given $x, x' \in R$, $f(x) - f(x')$ can also be written as the sum of a fixed part and a float part by substitution with Statement 1. We rearrange the equation so that the fixed part can be represented by the Jacobian matrix at $x$, while the float part can be viewed as a general instability caused by float neurons.

## 4.2 FLOAT PATH AND NETWORK ROBUSTNESS

Now that we have stated the motivation and properties of proposed concepts, the remaining question is how float and fixed paths affect the model robustness and randomized smoothing algorithms.

**Lipschitz Vulnerable.** We start with investigating the fixed part $J(x)(x - x')$, which is determined by both local Lipschitz constant and the scale of perturbation. Intuitively, given sphere $B(x, r)$, if the Lipschitz constant is larger enough, the fixed part in $f(x') - f(x)$ itself can alter the prediction of $x$. The threshold of $\|J(x)\|$ is determined by the sum of prediction margin $M(f(x), y) := min_{y' \neq y}|f(x)_y - f(x)'_y|$ and upper bound of float path $\mathcal{Z}^T(x', \mathcal{A}; B(x, r)) - \mathcal{Z}^T(x', \mathcal{A}'; B(x, r))$. We refer such an $x'$ as a **Lipschitz vulnerable data**. In particular, taking expectation on above sum, we have the following theorem.

**Theorem 2.** *Let $f$ be the base classifier. Given $(x, y) \sim D$ with $f(x) = y$, If*

$$\|J(x)\| > \frac{M(f(x), y) + \mathbb{E}[M(Z^T(x, \mathcal{A}; \mathcal{R}), y)]}{r},$$

*then for any smoothed classifier $g$ defined as above, there exist $x' \in B(x, r)$ such that $g(x') \neq y$.*

| $\eta$ | $\|\mathcal{Z}^I(x') + \mathcal{Z}^T(x')(1-\eta) - f(x)\|$ | | | | | Lipschitz Constant | Accuracy | | |
|---|---|---|---|---|---|---|---|---|---|
| | 0.05 | 0.10 | 0.15 | 0.20 | 0.25 | | $f(x)$ | $f(x_{adv})$ | $\mathcal{Z}^I(x_{adv})$ |
| Clean | 19.28 | 19.28 | 19.28 | 19.28 | 19.29 | 17.06 | 93.05 | 0.00 | 85.08 |
| $\sigma = 0.12$ | 17.14 | 16.84 | 16.57 | 16.33 | 16.12 | 4.99 | 86.25 | 12.54 | 65.90 |
| $\sigma = 0.25$ | 3.74 | 3.63 | 3.53 | 3.45 | 3.39 | 2.88 | 73.59 | 19.53 | 50.98 |

Table 1: Experiment results for VGG16 trained on CIFAR10 with noise at different scales.

The above theorem suggests that randomized smoothing fails to correct misclassified data $x$ if $x$ has an extremely large Lipschitz constant. Empirically, randomized smoothing cannot provide a certified radius for models trained with clean data, while can achieve minor accuracy on models trained with slightly perturbed data. The second set in Table 1 suggests that training models with noised sample can compress the Lipschitz constant therefore enables the effect of smoothed classifier.

**Float Neuron Vulnerable.** On the other hand, the float part contributes to $f(x') - f(x)$ differently. The last set in Table 1 presents the accuracy of FGSM (Goodfellow et al. (2014)) examples $x_{adv}$ with $\epsilon = 8/255$ under $l_\infty$ norm and the accuracy of sum of fixed paths $Z^I(x_{adv})$ between $\{x', x\}$. It shows that by removing the float paths between $x$ and $x'$, a significant accuracy boost can be achieved. This suggests that the unstable part of the network can greatly affect the prediction of network. We refer $x'$ as a float neuron vulnerable data if $\arg\max Z^I(x') = y$ while $f(x')$ is misclassified. Given $x'$ is float vulnerable, if the smoothed classifier corrected the prediction of $x'$, then the majority of the neighbor of $x'$ are voting for the correct label. In other word, the instability caused by float paths is smoothed by additional samples around $x'$.

**SC-RFP.** Above discussion suggests that smoothed classifier fails to boost the performance of Lipschitz vulnerable example, but is able to correct float neuron vulnerable data by smoothing the sudden change of float path in a region. Moreover, sampling more data shows that, the better smoothing is performed, the higher certified radius and accuracy can be achieved (Cohen et al. (2019)). In other words, randomized smoothing provides robustness by restricting the instability the network locally.

Based on this insight, SC-RFP introduces a manual repression on the local instability caused by float path into the smoothing classifier. The first set of Table 1 shows the averaged $l_1$ norm of difference between expectation of repressed prediction $\mathbb{E}[\mathcal{Z}^I(x') + (1 - \eta)\mathcal{Z}^T(x')]$ and $f(x)$, where $x'$ is perturbed by $\epsilon \sim N(0, 0.25)$ with $g(x') = y$. At each $x'$, we sample 2550 samples around $x'$ and compute the averaged distance to $f(x)$. We find that: (1) the clean model has high Lipschitz constant while repressing the instability from float path cannot reduce the prediction gap, and (2) for other models, repressing the instability drives the prediction of smoothed classifier towards original prediction $f(x)$. The results support above discussion. Algorithm 1 describes SC-RFP in details[3].

At each block, we first compute the pre-activation value of $x$ and $x + \epsilon$. Prior to passes to the activation function, we compare the activation pattern between $z_i(x)$ and $z_i(x + \epsilon)$ according to the separation $\Gamma$, and use a repression factor $\eta$ to reduce the value of float path.

---

**Algorithm 1** Smoothed Classifier with Repressed Float Path
**Inputs:** Network $\mathcal{N}$ with parameter $\theta$, randomized algorithm $g$, input $x$, repress ratio $\eta$.
**Outputs:** Predicted label $\hat{y}$

---

  **while** $g$ sample noise $\epsilon$ **do**
    **for** Block $i$ in Block $1, 2, \ldots, d - 1$ **do**
      $\mathcal{I}_i^T \leftarrow \mathcal{A}(z_i(x)) \neq \mathcal{A}(z_i(x + \epsilon))$
      $z_i(x + \epsilon) \leftarrow z_i(x + \epsilon) - z_i(x + \epsilon) \times \mathcal{I}_i^T \times \eta$
      $x_{i+1}(x + \epsilon) \leftarrow \sigma(z_i(x + \epsilon))$
    **end for**
    $counts \leftarrow \phi_d(x_d(x + \epsilon))$
  **end while**
  $\hat{c}_A, \hat{c}_B \leftarrow$ top two indices in $counts$
  $\hat{n}_A, \hat{n}_B \leftarrow counts[\hat{c}_A], counts[\hat{c}_B]$
  **if** $BinomPValue(n_A, n_A + n_B, 0.5) \leq \alpha$ **then** return $\hat{c}_A$ **else** return Abstain
  **end if**

---

Intuitively, manipulating the computational graph can greatly change the prediction result, while we show that there are only a small proportion of path are affected by our method. Figure 2 present the proportion of fixed neuron between $x$ and $x + \epsilon$ of VGG16 network trained on CIFAR10. It shows that models trained with noised sample have relatively more stable activation pattern, as the ratio of fixed neurons are lower. In particular, the model trained with clean data has average fixed ratio around 72%, while after adding a minimum scale of noise $\sigma = 0.05$, it increases to around 90%.

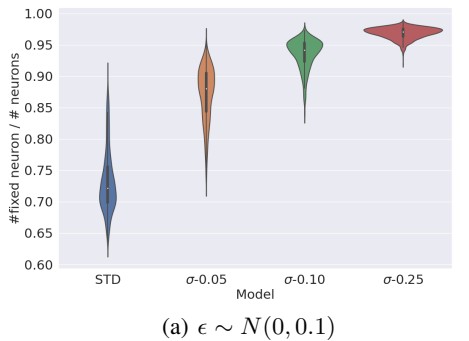
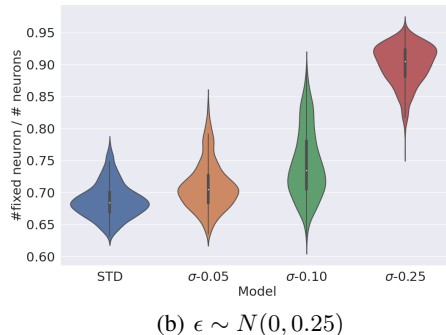

(a) $\epsilon \sim N(0, 0.1)$          (b) $\epsilon \sim N(0, 0.25)$

Figure 2: Proportion of fixed neuron between $x$ and $x + \epsilon$ for VGG16 models trained on CIFAR10 with noised data at different scales: clean data, $\sigma = 0.05$, $\sigma = 0.10$ and $\sigma = 0.25$. Figure 2(a) and 2(b) show the ratio given $\epsilon \sim N(0, 0.1)$ and $\epsilon \sim N(0, 0.25)$.

At the end of this section, we present Theorem 3 to link the certifiable boundary of proposed algorithm with previous works.

---

[3]Code is provided at: https://github.com/OrangeBai/APCT-master

**Theorem 3.** *Let $\mathcal{N}$ be a network defined as Section 3.1. Let $g$ be a smoothed classifier that samples noise from distribution $D_{noise}$ and $g'$ is the SC-RFP built on $g$. Given $\epsilon \sim D_{noise}$, assume that the direction of $\epsilon$ is uniformly distributed:*

$$\forall \|\eta_1\| = \|\eta_2\| = 1, P(\frac{\epsilon}{\|\epsilon\|} = \eta_1) = P(\frac{\epsilon}{\|\epsilon\|} = \eta_2). \tag{10}$$

*If $\arg\max_{m \in Y} f_y(x) = y$, then*

$$\underline{p'_A} > \underline{p_A}, \overline{p_B}' < \overline{p_B}, \tag{11}$$

*where $\underline{p'_A}, \underline{p_A}$ are the lower bound of $g'_y(x)$ and $g_y(x)$, $\overline{p_B}', \overline{p_B}$ are the upper bound of $g'_{m \neq y}(x)$ and $g'_{m \neq y}(x)$. Moreover, if $\arg\max_{m \in Y} g(x) = y$, $g'(x)$ has certified radius no less than $R$, where*

$$R = \frac{\sigma}{2}(\Phi^{-1}(\underline{p_A}) - \Phi^{-1}(\overline{p_B})). \tag{12}$$

Notice that in the above theorem, we do not specify the randomized classifier $g$ but introduce a constraint of the randomness of $g$. This means that the theorem holds for not only the naive smoothed classifier, but also its variants. However, it does not hold for the classifiers that sample directed noise.

## 5 EXPERIMENTS

As SC-RFP is built on a base smoothed classifier, we measure the method on different model structures and randomized smoothing algorithms. The experiments are performed on CIFAR10 and ImageNet datasets. For the CIFAR10 dataset, we compare our method with the benchmark classifier proposed by Cohen et al. (2019) with VGG16 network to show the effectiveness of our method. For the ImageNet dataset, we choose ResNet50 as model architecture and add the adversarial smoothed classifier Salman et al. (2019) as the base model. We also compare our model with recent works (Jeong & Shin (2020); Jeong et al. (2021)) to obtain a general evaluation. In line with previous works, we use certifiable accuracy at different radius computed by Cohen et al. (2019) as the metric.

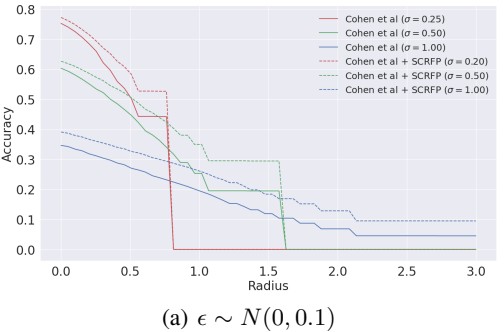

(a) $\epsilon \sim N(0, 0.1)$

Figure 3: Certified Accuracy of base methods and SC-RFP ($\eta = 0.1$) with different level of noise on CIFAR10. The solid and dashed lines represent benchmark and SC-RFP.

**CIFAR10.** We train the VGG16 network with different scales of noisy samples. Each of the models is trained for 200 epochs with SGD optimizer and an initial learning rate of 0.1, which decays after 60, 120, and 160 epochs with a rate of 0.2. Table 2 compares the robust accuracy of SC-RFP and benchmark. Generally, our SC-RFP increases have an increasingly robust accuracy between 6% to 10% compared with the benchmark. We notice that by repressing the float path, SC-RFP is able to improve the accuracy of clean data, which supports our analysis that the float neuron vulnerability is one of the causes of misclassification. Figure 3 present the robust accuracy at different noise scale. We find that the performance boost from SC-RFP increases as the radius increases for all the models.

| Method | 0.25 | 0.5 | 0.75 | 1.0 | 1.25 | 1.5 | 2.0 | 2.5 | Clean |
|---|---|---|---|---|---|---|---|---|---|
| Cohen et al. (2019) | 66.0 | 50.2 | 44.2 | 25.4 | 19.5 | 19.5 | 6.9 | 4.5 | 75.3 |
| + SC-RFP | **69.4** | **58.3** | **52.7** | **34.9** | **29.5** | **29.5** | **12.9** | **9.5** | **77.2** |

Table 2: Certified robust accuracy for Benchmark and SC-RFP ($\eta = 0.1$) on CIFAR10.

**Experiments on ImageNet.** We train ResNet50 using the benchmark model (Cohen et al. (2019)) and SmoothAdv (Salman et al. (2019)) on ImageNet-2012 for 80 epochs with same training scheduler and optimizers as the original work. Each of the model are certified with same level of noise as they were trained.

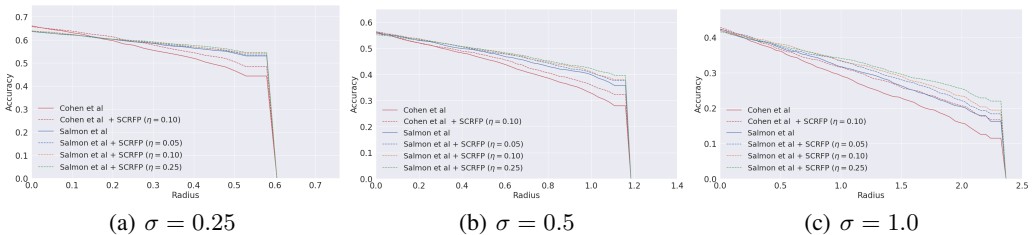

(a) $\sigma = 0.25$          (b) $\sigma = 0.5$          (c) $\sigma = 1.0$

Figure 4: Certified Accuracy of base methods and SC-RFP on ImageNet. Model are trained and tested with (a) $\sigma = 0.25$, (b) $\sigma = 0.5$ and (c) $\sigma = 1.0$. The solid and dashed lines represent benchmark and SC-RFP with different repression factor.

Table 3 compares the certified robust accuracy of different models at different radius on ImageNet. It shows that when deployed on the PGD+noise training (Salman et al. (2019)), the SC-RFP has the highest robust accuracy at all the radius. Moreover, an increasing boost in the robust accuracy is observed as the radius increase, which is consistent with our previous analysis of CIFAR10 result.

Figure 4 provides more details on our experiments. For a noise level of $\sigma = 0.25$, SmoothAdv and SmoothAdv + SC-RFP has neglectable difference, while in Figure 4(c), the SC-RFP ($\sigma = 0.25$) increases the robust accuracy by around 5%. By introducing SC-RFP ($\sigma = 0.1$) to the benchmark classifier, it achieves comparable results with other state-of-art models when $l_2$ radius is around 1.5.

| $l_2$ radius | 0.25 | 0.5 | 0.75 | 1.0 | 1.5 | 2.0 | Clean |
|---|---|---|---|---|---|---|---|
| Jeong & Shin (2020) | 59.8 | 49.8 | 44.7 | 39.3 | 28.1 | 22.6 | - |
| Jeong et al. (2021) | 46.7 | 38.2 | 30.3 | 26.8 | 15.7 | 12.1 | - |
| Cohen et al. (2019) | 57.9 | 46.4 | 41.9 | 32.6 | 22.8 | 15.6 | **66.1** |
| + SC-RFP (ours) | 59.4 | 50.3 | 41.5 | 35.6 | 26.1 | 20.2 | 65.8 |
| Salman et al. (2019) | 59.2 | 54.1 | 51.5 | 39.6 | 26.2 | 20.1 | 63.6 |
| + SC-RFP (ours) | **59.8** | **55.4** | **53.6** | **42.0** | **29.6** | **25.2** | 63.9 |

Table 3: Certified robust accuracy for models with different methods on ImageNet

We find two insightful observations from the above results. First, introducing the repression factor to the float path has a minimum negative effect on the standard accuracy, while can greatly boost the robust accuracy. This supports our discussion that float neuron vulnerable examples are caused by the float path, and can be cured by reducing their contribution. Second, we find that as the radius grows, the boost from our method increases. This is consistent with the theoretical basis of SC-RFP as well as the result from CIFAR10 dataset. When size of the perturbation grows, the uncertainty of the randomized algorithm increases along with the number of float paths as Figure 2 suggests. This results in a higher abstain rate and a higher possibility of misclassification. By repressing the value of the float path, the deviation of $g(x + \epsilon)$ from $f(x)$ is reduced, therefore a more stable result can be achieved.

## 6 CONCLUSION

In this work, we introduce SC-RFP algorithm that can improve the performance of a randomized smoothing classifier. We first introduce a framework for describing the local activation status of neurons and show that most of the neurons are locally stable, while the others can greatly affect the model prediction. By decomposing the computational graph of the network, we find that the boost of robust accuracy provided by the smoothed classifier is averaging the deviation of float paths. Based on this, we suggest further repressing the value of float paths with SC-RFP method. The experiments show that our method can improve the performance of smoothing models.

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

## APPENDIX

The Appendix contains four sections. Section A further discusses the geometric intuition following Section 3.3. Section B provides proofs of the decomposition of computational graph following Section 4. Section C further invesitgates the properties of proposed SC-RFP algorithm. At last, Section D provide supplementary experiment results for Section 5

## A  GEOMETRIC ILLUSTRATION

This section provides geometric illustration of the proposed framework. For the completeness of definition of activation region, we discuss the case that pre-activation value $z_{ij}(x)$ locates within none of the intervals. In other words, we consider the case that $z_{ij}(x; \theta) \in \{\gamma_1, \gamma_2, \ldots, \gamma_q\}$.

**Definition 1** (Generalized Activation Pattern / Region (Restate))**.** *Let $\mathcal{N}$ be a network defined as in Section 3.1. Denote*
$$\Gamma = \{\gamma_1, \gamma_2, \ldots, \gamma_q\}$$
*as a set of breakpoints that separates the domain of activation function into $q + 1$ intervals*
$$U = \{U_0, U_1, \ldots, U_q\}.$$
*An activation pattern of $\mathcal{N}$ is defined as an indexed family that assigns each neuron a sign to represent the status of its pre-activation in $U$:*
$$\mathcal{A} := \{a_{ij} \, | a_{ij} \in \{0, 1, \ldots, q\}, (i, j) \in \mathcal{I}\}.$$
*Given an activation pattern $\mathcal{A}$, the activation region of $\mathcal{A}$ is defined as:*
$$\mathcal{R}(\mathcal{A}; \theta, \sigma, \Gamma) := \{x \in R^{in} | z_{ij}(x; \theta) \in U_{a_{ij}}, a_{ij} \in \mathcal{A}\} \tag{13}$$
*Conversely, given $x \in R^{in}$, we denote the activation pattern of neuron $(i, j)$ at $x$ as:*
$$\hat{a}_{ij}(x; \theta, \sigma, \Gamma) = \begin{cases} i, & z_{ij}(x; \theta) \in U_i, \\ abstain, & z_{ij}(x; \theta) \in \{\gamma_1, \gamma_2, \ldots, \gamma_q\} \end{cases} \tag{14}$$
*The activation pattern of $x$ is the collection of the all the pattern of neurons:*
$$\hat{\mathcal{A}}(x; \theta, \sigma, \Gamma) = \{\hat{a}_{ij}(x) | (i, j) \in \mathcal{I}\} \tag{15}$$

**Inverse of activation region operator.** The generalized activation pattern describes the activation status of each unit at the intermediate layers. Given an activation pattern, $\mathcal{R}(\cdot)$ finds the region, such that, for every $x \in \mathcal{R}(\mathcal{A}; \theta, \sigma, \Gamma)$, the pre-activation value of each neuron $z_{ij}(x; \theta)$ locates within the interval $U_{a_{ij}}$ that determined by its pattern $a_{ij} \in \{0, 1, \ldots, q\}$. However, the activation pattern of $x$ given $z_{ij}(x) \in \{\gamma_1, \gamma_2, \ldots, \gamma_q\}$ cannot be determined under previous definition as well as previous works(Hanin & Rolnick (2019b)Raghu et al. (2017)). Therefore, we introduce an inverse operator $\hat{\mathcal{A}}(x; \theta, \sigma, \Gamma)$ that computes the activation pattern of an input $x \in R^{n_0}$ in Definition 1. This allows us to fill the completeness with the framework.

Next, we consider the set $\{x | z_{ij}(x) \in \{\gamma_1, \gamma_2, \ldots, \gamma_q\}\}$. Under trivial assumption that the probability distribution of parameter set $\{\theta\}$ has no atom:
$$\mathbb{P}(\theta = \theta_0) = 0, \forall \theta_0 \in R, \theta \in \{theta\}.$$

The Lebesgue measure of $\{x|z_{ij}(x) \in \{\gamma_1, \gamma_2, \ldots, \gamma_q\}\}$ is 0 in $R^{n_0}-$ dimension. From a geometric point of view, the collection of those $x$ partition the input space $R^{n_0}$ into numerous regions by a set of hyperplanes $\{H_{ijk}(\theta)\}$, where:

$$H_{ijk}(\theta) := \{x \in U|z_{ij}(x;\theta) = \gamma_k\},$$

is determined by neuron $(i, j)$ and breakpoint $\gamma_k$. Each of the region, as Definition 1, is an activation region. The following figure presents a geometric illustration similar to that in the Section 4, but with the description of bent-hyperplanes.

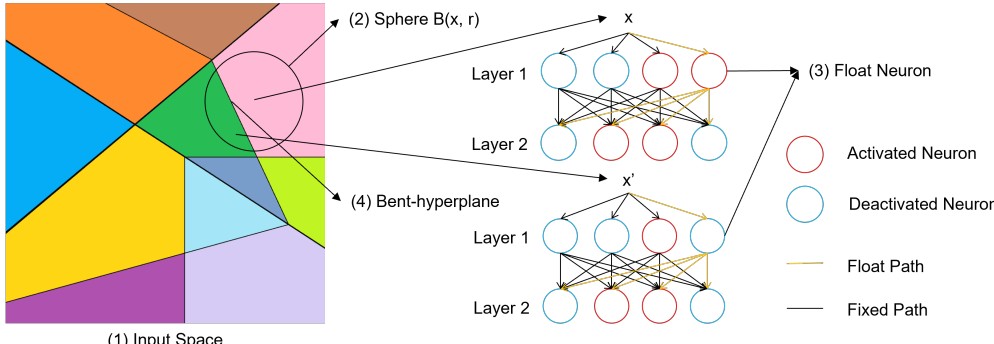

Figure 5: An illustration of the fixed (float) path and neuron of a neural network with 2D input, 4D output, and 1 hidden layer. (1) The 2D input space; (3) A sphere centered at $x$; (4) A float neuron with index (1,4) in the network.; (4) A bent-hyperplane defined by $H = x|z_{1,4}(x) = 0$;

Figure 5 presents an illustration of the key concepts of this work. Consider $N$ as a neural network with 2D input, 4D output and 1 hidden layer with 4 neurons. Assume that $N$ has ReLU activation function. The input space is partitioned into several regions by a set of hyperplanes. Each of the region is referred as an activation region that marked in a unique color, within which the mapping function is linear.

For example, the pink and green region are divided by the bent-hpyerplane $H = x|z_{1,4}(x) = 0$, where $z_{1,4}$ is the pre-activation value of 4-th neuron in the 1st layer. Now we consider a sphere $B(x, r)$ centered at $x$ with radius $r$. $B(x, r)$ is covered by the union of pink and green region. Since the two regions can be merged into one by removing the hyperplane, then the incomplete activation pattern of the merged region is $\mathcal{I}\setminus\{(1, 4)\}$. Therefore, $(1, 4)$ is the only float neuron in $B(x, r)$. This means that for every $x \in B(x, r)$, all the neuron has same activation pattern expect for the neuron $(1, 4)$.

Now we consider the computational path of $x \in B(x, r)$. Given a path $\zeta$, if neuron $(1, 4)$ is not on this path, then $\zeta$ is a fixed path, which are the black lines in the figure. This means that for every $x \in B(x, r)$, the value of those fixed path are linear function with respect to $x$. On the other hand, all the non-linearity of function $f(x)$ for $x \in B(x, r)$ are contributed by the float path (Orange lines).

With the illustration above, we proof Lemma 1 as follows.

**Lemma 1** (Restated). *Let $N$ be a neural network defined as Section 3.1. Given $R \subset R^{n_0}$, denote $\mathcal{I}^X$ and $\mathcal{I}^T$ are the set of float neuron and fixed neuron in R. Then:*

    *1. $\mathcal{I}^X(R)\bigcup\mathcal{I}^T(R) = \mathcal{I}$*            *2. $R \subset \mathcal{R}(\mathcal{A}_{\mathcal{I}^X})$*

**Proof.** *Given a neuron $(i, j)$ and a region $R \in R^{n_0}$. For any $x \in R$, we use $\hat{a}_{ij}(x)$. Since $z_{ij}(x) \in (-\infty, \infty)$, the pattern of $x$ is either an index $k$ of the region when $z_{ij}(x) \in U_k$ or -1 when $z_{ij}(x)$ locate on the bent-hyperplane $H_{ijk} := \{x|z_{ij}(x) = k\}$.*

*This means every neuron has a pattern for $x \in R$. If for every $x \in R^{n_0}$, the pattern of neuron $(i, j)$ remains the same, then $(i, j)$ is a fixed neuron. Otherwise, it is a float neuron. In other words, a neuron is either fixed or float in $\mathbb{R}$. This implies that $\mathcal{I}^X(R)\bigcup\mathcal{I}^T(R) = \mathcal{I}$.*

*Now we consider statement 2. For any $x_1, x_2 \in R$, and an index of neuron $(i, j)$. If $\hat{a}_{ij}(x_1) = \hat{a}_{ij}(x_2)$, $(i, j)$ is a fixed neuron in R: $(i, j) \in I^X$. Denote the pattern $a_i j$. Then for all $x_1 \in R$, $z_{ij;x_1,\theta} = z_{ij;x_2,\theta}$. Therefore, $x_1 \in \mathcal{R}(\mathcal{A}^X)$.*

## B   DECOMPOSING THE NETWORK

Before we delve into details, we have to restate the definition of path and introduce definition of sub-path. This enables us to take the bias, the computational graph of which starts at middle of the network instead of the beginning, into account. We start with introducing a sub-path of network $\mathcal{N}$.

**Definition 6** (Sub-path). *A sub-path of a path consists of several consecutive elements of $\gamma$:*

$$\zeta_{(i,j)} = \{\zeta_j, \zeta_{j+1} \ldots, \zeta_i\} \subseteq \zeta \tag{16}$$

*We use $\zeta_{(i,j)}(x)$ to represent the computation graph from $(j, \zeta_j)$ to $(i, \zeta_i)$:*

$$\zeta_i(x) = \zeta_{(i,j)} \circ \zeta_{j,0}(x) \tag{17}$$

**Definition 4** (Path (Restate)). *Let $\mathcal{N}$ be a network defined as Section 3.1. A path of $\mathcal{N}$ is a set of neurons:*

$$\zeta := \{(i, \zeta_i) | i = 0, 1, \ldots, d\}, \zeta_i \in \{1, 2, ..., n_i\}, \tag{18}$$

*where $n_i$ is the number of neurons in layer $i$. The value of a path is defined as:*

$$\zeta_{(i,j)}(v, \mathcal{A}) := v \prod_{m=j+1}^{i} d_{\mathcal{A}_{m,\varsigma_m}} W'^{(m)}_{(\zeta_m, \zeta_{m-1})}, \tag{19}$$

*where $x_{\zeta_j}$ is the $\zeta_j$-th element of input, $W'^{(i)}$ is the equivalent matrix linear transformation $\psi_i \circ \phi_i$, $a_{m,\varsigma_m}$ is the activation pattern of neuron $(m, \varsigma_m)$ and $d_{a_{m,\varsigma_m}}$ is the slope of activation $\sigma$ within $a_{m,\varsigma_m}$-th interval.*

Next, we consider the mapping from $v$, which can be either from $x$ or a bias at intermediate layers, to output of $i$-th layer $y_i$.

**Lemma 2.** *Let $\mathcal{N}$ be a neural network defined as Section 3.1. Given activation pattern $\mathcal{A}$, for any $x \in \mathbb{R}(\mathcal{A}; \theta, \sigma, \Gamma)$. The $k$-th component of output vector of layer $i$ can be represented as:*

$$y_{ik}(x) = \sum_{\forall \zeta_i=k} x_{j,\zeta_j} \prod_{m=j+1}^{i} d_{\mathcal{A}_{m,\varsigma_m}} W'^{(m)}_{(\zeta_m, \zeta_{m-1})} + \sum_{o=j}^{i-1} \sum_{\forall \zeta_i=k} \beta^{(o)}_{\zeta_o} \prod_{m=o+1}^{i} d_{\mathcal{A}_{m,\varsigma_m}} W'^{(m)}_{(\zeta_m, \zeta_{m-1})} \tag{20}$$

*where $x_{j,\zeta_j}$ is the $\zeta_j$-th element of input in layer $j$, $W'^{(i)}$ and $\beta^{(i)}$ are the equivalent matrix and bias of linear transformation $\psi_i \circ \phi_i$, $\mathcal{A}_{m,\varsigma_m}$ is the activation pattern of $\zeta_m$-th component of layer $m$ and $d_{\mathcal{A}_{m,\varsigma_m}}$ is the slope of pattern $\mathcal{A}_{m,\varsigma_m}$: $d_{\mathcal{A}_{m,\varsigma_m}} := \sigma'(t), t \in U_{\mathcal{A}_{m,\varsigma_m}}$.*

**Proof** (Lemma 2). *We first show that the pre-activation transformation can be represented by a matrix. As $\phi_i$ is a linear affine, we denote $\phi_i(x) = W_i x_i$. Combing with batch normalization layer, the mapping from input $x_i$ to pre-activation is:*

$$z_i = \gamma_i \frac{\phi_i(x) - \hat{\mu}}{\hat{\sigma}} + \beta_i = \frac{\gamma}{\hat{\sigma}} \phi_i(x) - \frac{\gamma \hat{\mu}}{\hat{\sigma}} + \beta_i = D_i(\frac{\gamma}{\hat{\sigma}}) W_i x - \frac{\gamma \hat{\mu}}{\hat{\sigma}} + \beta_i \tag{21}$$

*where $D_i(\cdot)$ is the diagonal operator, $\hat{\mu}$ and $\hat{\sigma}$ are the mean and variance parameter of the $\psi_i$. Denote $W'^{(i)} = D_i(\frac{\gamma}{\sigma}) W_i$, $\beta'_i = (\beta_i - \frac{\gamma \hat{\mu}}{\hat{\sigma}})/n_i$. We proof this byh deduction. For $i = 1$:*

$$z_i = \sum_{j=0}^{n_0} W'^{(1)}_{ij} d_{\mathcal{A}_{1,i}} x_{0j} + n_1 \times \beta'^{(1)}_i.$$

*where $\mathcal{A}_{1,i}$ is the activation pattern of $z_{1i}$. Since all the path in layer $i$ ends at $i$ are $\{(1, i), (2, i), \ldots, (n_0, i)\}$. Equation 20 holds.*

*Now we assume equation holds for $i = p$.*

$$
\begin{aligned}
z_{pi} &= \sum_{j=0}^{n_i} W'^{(i)} x_{ij} + n_i \times \beta_j'^{(i)} \\
&= \sum_{j=0}^{n_i} W'^{(i)} z_{i-1,j} + n_i \times \beta_j'^{(i)} \\
&= \sum_{j=0}^{n_i} W'^{(i)} d_{\mathcal{A}_{i,j}} z_{i-1,j} + n_i \times \beta_j'^{(i)} \\
&= \sum_{j=0}^{n_i} W'^{(i)} d_{\mathcal{A}_{i,j}} \big( \sum_{\forall \zeta_i = j} x_{j,\zeta_j} \prod_{m=p}^{i} d_{\mathcal{A}_{m,\zeta_m}} W'^{(m)}_{(\zeta_m,\zeta_{m-1})} \\
&\quad + \sum_{o=p-1}^{i-1} \sum_{\forall \zeta_i = k} \beta_{\zeta_o}^{(o)} \prod_{m=o+1}^{i} d_{\mathcal{A}_{m,\zeta_m}} W'^{(m)}_{(\zeta_m,\zeta_{m-1})} \big) + \beta_j'^{(i)} \\
&= \sum_{\forall \zeta_p = k} x_{j,\zeta_j} \prod_{m=j+1}^{p} d_{\mathcal{A}_{m,\zeta_m}} W'^{(m)}_{(\zeta_m,\zeta_{m-1})} + \sum_{o=j}^{p-1} \sum_{\forall \zeta_p = k} \beta_{\zeta_o}^{(o)} \prod_{m=o+1}^{i} d_{\mathcal{A}_{m,\zeta_m}} W'^{(m)}_{(\zeta_m,\zeta_{m-1})}
\end{aligned}
\tag{22}
$$

Lemma 2 decomposes the computational graph of network $N$, while Theorem 1 categorize the paths of $f(x)$ into fixed part and float part. Moreover, it describes the model robustness with by divide the variant of $f$ into a linear part as well as a non-linear part.

**Theorem 1** (Restate). *Let $N$ be a neural network defined as 3. Given $R \subset X$, for any $x, x' \in R$ with activation pattern $\mathcal{A}$ and $\mathcal{A}'$, we have :*

1. *$f(x) = \mathcal{Z}^I(x, \mathcal{A}; \mathcal{R}) + \mathcal{Z}^T(x, \mathcal{A}; \mathcal{R})$*

2. *$f(x) - f(x') = J(x)(x - x') + \mathcal{Z}^T(x', \mathcal{A}; \mathcal{R}) - \mathcal{Z}^T(x', \mathcal{A}'; \mathcal{R})$*

*where $\mathcal{Z}^I(x, \mathcal{A}; \mathcal{R}) = \sum_{\zeta \in \mathcal{Z}^I(\mathcal{R})} \zeta(x, \mathcal{A})$, $\mathcal{Z}^T(x, \mathcal{A}; \mathcal{R}) = \sum_{\zeta \in \mathcal{Z}^T(\mathcal{R})} \zeta(x, \mathcal{A})$ are the sum of fixed path and float path given the region $\mathcal{R}$, $J(x)$ is the Jacobian matrix of $f$ at $x$.*

Before we proof above Theorem, we use the following Lemma to show that every fixed path in a region is linear regardless of the activation pattern.

**Lemma 3.** *Let $N$ be a network defined as Section 3.3. Let $\zeta$ be a fixed path in $R \subset R_{n_0}$. Then for any $x, x' \in R$ with activation pattern $\mathcal{A}, \mathcal{A}'$, $\zeta(x', \mathcal{A}) = \zeta(x', \mathcal{A}')$. Moreover, $\mathcal{Z}^I(x', \mathcal{A}; \mathcal{R}) = \mathcal{Z}^I(x', \mathcal{A}'; \mathcal{R})$.*

**Proof** (Lemma 3). *Given $x \in R$ with activation pattern $A$,*

$$
\zeta(x, \mathcal{A}) := v \prod_{m=1}^{d} d_{\mathcal{A}_{m,\zeta_m}} W'^{(m)}_{(\zeta_m,\zeta_{m-1})}.
$$

*Since $\zeta$ is a fixed path in $R$, then every neuron alone $\zeta$ is fixed neuron, the activation pattern of $(m, \zeta_m)$ is same regardless of input $x$. Therefore, $d_{\mathcal{A}_{m,\zeta_m}}$ is constant for any $x \in R$. Then $\zeta(x, \mathcal{A})$ is a linear function of $x : \zeta(x, \mathcal{A}) = \zeta(x)$. In other words, the change of activation pattern does not affect the neurons on $\zeta$, therefore the slope of this path does not change. $\zeta(x)$ is dependent on $x$ in region $R$. This means that $\zeta(x', \mathcal{A}) = \zeta(x', \mathcal{A}')$ for any $x, x' \in R$ with activation pattern $\mathcal{A}, \mathcal{A}'$*

*$\mathcal{Z}^I(x', \mathcal{A}; \mathcal{R})$ is the aggregation of all the fixed path above. Since the summation of linear function is still linear, we have:*

$$
\mathcal{Z}^I(x', \mathcal{A}; \mathcal{R}) = \mathcal{Z}^I(x', \mathcal{A}'; \mathcal{R})
$$

**Proof** (Theorem 1). *From Lemma 1, every neuron is either fixed neuron or float neuron. For the neurons in path $\zeta$, if there exists a float neuron, then the path is float path. Otherwise, it is a fixed path. The float path and fixed path are complementary set on the set of all paths.*

*Lemma 2 decomposes the $f(x)$ into summation of paths. As each path is either float or fixed, we have*

$$
f(x) = \mathcal{Z}^I(x, \mathcal{A}; \mathcal{R}) + \mathcal{Z}^T(x, \mathcal{A}; \mathcal{R}).
$$

*For statement 2, we have:*

$$f(x) - f(x') = \left(\mathcal{Z}^I(x, \mathcal{A}; \mathcal{R}) + \mathcal{Z}^T(x, \mathcal{A}; \mathcal{R})\right) - \left((\mathcal{Z}^I(x', \mathcal{A}'; \mathcal{R}) + \mathcal{Z}^T(x', \mathcal{A}'; \mathcal{R})\right)$$
$$= \left(\mathcal{Z}^I(x, \mathcal{A}; \mathcal{R}) + \mathcal{Z}^T(x, \mathcal{A}; \mathcal{R}) - \mathcal{Z}^I(x', \mathcal{A}; \mathcal{R}) - \mathcal{Z}^T(x', \mathcal{A}; \mathcal{R})\right) + \quad (23)$$
$$\mathcal{Z}^T(x', \mathcal{A}; \mathcal{R}) - \mathcal{Z}^T(x', \mathcal{A}; \mathcal{R})$$

*Notice that, $\mathcal{Z}^I(x', \mathcal{A}; \mathcal{R})$ is the collection of all fixed path, therefore is unrelated with the change of $\mathcal{A}$. We have $\mathcal{Z}^I(x', \mathcal{A}; \mathcal{R}) = \mathcal{Z}^I(x', \mathcal{A}'; \mathcal{R})$. The former part of above equation is equals to $J(x)(x - x')$, which reaches to statement 4.*

## C  MODEL ROBUSTNESS

### C.1  LIPSCHITZ VULNERABLE AND FLOAT NEURON VULNERABLE

In Section 4.2 we discuss two kinds of vulnerable data and discuss our motivation of SC-RFP. In this section, we present formally statements of the proposed idea.

Before we move to Theorem 2, we need the following two lemmas. Lemma 5 suggests that, given a fixed path $\zeta$, the expectation for the randomized algorithm $g(\zeta(x))$ is equal to the $\zeta(x)$ when $g$ satisfies certain conditions. In fact, the condition here just assume that the randomly sampled noise has equal probability at each direction, which is a natural assumption for most sampling procedure.

**Lemma 4.** *Denote $N$ as a neural network as Definition 3 with mapping function $f$, $\zeta$ is a fixed path in $B_2(x, p)$. Assume $g$ is a randomized algorithm that applies noise $\epsilon$ from certain distribution $D_{noise}$ on the computation path of $\zeta$:*

$$g(\zeta(x)) = \zeta_{d,i}(\zeta_{i,0}(x) + \epsilon), i \in \{0, 1, \ldots, d-1\}, \epsilon \sim D_{noise} \quad (24)$$

*If the direction of $\epsilon$ is uniformly distributed, then the expectation of $g(\zeta(x))$ is equals to $\zeta(x)$, that is:*

$$\forall \|\eta_1\| = \|\eta_2\| = 1, P(\frac{\epsilon}{\|\epsilon\|} = \eta_1) = P(\frac{\epsilon}{\|\epsilon\|} = \eta_2) \Rightarrow E[g(\zeta(x))] = \zeta(x), \quad (25)$$

*where $\epsilon$ is the noise generated for randomized algorithm $g$.*

**Proof.** *Assume that $g$ samples noise at layer $k$. Then for any $g$, $\zeta(k, 0) = g(\zeta_{k,0})$, which we denote as $x_k$. Since the activation function is piecewise linear, the mapping of $\zeta_{d,k}(\cdot)$ is linear. We have:*

$$\sum_i^{n_k} \frac{\partial^2 y_j}{\partial z_{kj}^2} = 0. \quad (26)$$

*$\zeta_{d,k}$ is harmonic. Therefore, given any $B(x, r)$ with radius $r > 0$:*

$$\zeta_{d,k}(x) = \frac{1}{n\omega_n r^{n-1}} \int_{\partial B(x,r)} \zeta d\sigma, \quad (27)$$

*where $\omega_n$ is the volume of the unit ball in $n$ dimensions, $\sigma$ is the $(n-1)$ dimensional surface measure. Now we consider the expectation of $g(x)$:*

$$\mathbb{E}(g(x)) = \int p(\epsilon)\zeta_{d,k}(x_k + \epsilon)d\mu$$
$$= \int \int_{\|\epsilon\|=r} p(\epsilon)\zeta_{d,k}(x_k + \epsilon)d\mu dr, \quad (28)$$

*where $\mu$ is the probability measure of $\epsilon$ $D_{noise}$. As the noise sampling is assumed to be direction irrelevant, the measure of $p(\epsilon|\|\epsilon\|_2 = r)$ is uniformly distributed given radius $r$. Therefore:*

$$\int_{\|\epsilon\|=r} p(\epsilon)\zeta_{d,k}d\mu = P(\|\epsilon\|_2 = r)\zeta_{d,k}. \quad (29)$$

*Equation 28 then equals:*

$$\mathbb{E}(g(x)) = \int p(\epsilon)\zeta_{d,k}(x_k + \epsilon)d\mu$$
$$= \int P(\|\epsilon\|_2 = r)\zeta_{d,k}dr \quad (30)$$
$$= \zeta_{d,k}(x)$$

**Lemma 5.** *Let $f$ and $g$ be the mapping function and randomized classifier defined as above. Given radius $r$ such that, almost surely, $\mathbb{Z}^T(B(x,r)) = \emptyset, \forall (x,y) \sim D$, then the accuracy of base classifier and the naive smoothed classifier are same, that is:*

$$\mathbb{E}_{(x,y)\sim D} \left[ \arg\max_{m\in Y} g_m(x) = y \right] = \mathbb{E}_{(x,y)\sim D} \left[ \arg\max_{m\in Y} f_m(x) = y \right]$$

Lemma 5 is the achieved by directly applying Lemma 4 on the all the computational graph of the network. It shows that if all the neurons have locally stable activation pattern with respect to the distribution of dataset, the smoothed classifier provides identical accuracy with the base classifier.

Next, we present the proof of Theorem 2.

**Theorem 2.** *Let $f$ be the base classifier. Given $(x,y) \sim D$ with $f(x) = y$, denote $M(f(x),y) := min_{y'\neq y}|f(x)_y - f(x)'_y|$ as the margin operator of prediction vector. If*

$$\|J(x)\| > \frac{M(f(x),y) + \mathbb{E}[M(Z^T(x,\mathcal{A};\mathcal{R}),y)]}{r},$$

*then for any smoothed classifier $g$ defined as above, there exist $x' \in B(x,r)$ such that $g(x') \neq y$.*

**Proof.** *Statement 4 of 1 suggests:*

$$f(x) - f(x') = J(x)(x - x') + \mathcal{Z}^T(x',\mathcal{A};\mathcal{R}) - \mathcal{Z}^T(x',\mathcal{A}';\mathcal{R}) \tag{31}$$

*As $\|J(x)\| > \frac{M(f(x),y) + \mathcal{M}(\mathbb{E}[Z^T(x,\mathcal{A};\mathcal{R})],y)}{r}$, there exist $x'$ and $i$ such that:*

$$f_i(x') > f_y(x') + \mathbb{E}[Z_y^T(x,\mathcal{A};\mathcal{R}) - Z_i^T(x,\mathcal{A};\mathcal{R})] \tag{32}$$

*Therefore,*

$$g_i(x) - g_i(x') > g_y(x) - g_y(x') + M(f(x),y),$$
$$g_i(x') > g_y \tag{33}$$

Next, we consider the case that $x$ is an adversarial example that mislead $f$ but correctly classified by $g$, which we referred to as a float neuron vulnerable example. The following theorem suggests that, the float path at $x$ are the cause of the altered prediction. In other words, the network is locally correct around $x$, while there are sudden change from the float path that causes the misclassification.

**Theorem 4.** *Let $f$ and $g$ be the base and smoothed classifier defined above. Given $(x,y) \sim D$, denote $x' \in B(x,r)$ as an adversarial example that mislead the base classifier but corrected by $g$ without abstain:*

$$\arg\max_{m\in Y} f_m(x) = i, \quad \arg\max_{m\in Y} g_m(x) = y, i \neq y \tag{34}$$

*then loss of float path is higher than that of $g$:*

$$CE(\mathcal{Z}^T(x',\mathcal{A};\mathcal{R}), onehot(y)) > CE(g(\mathcal{Z}^T(x',\mathcal{A};\mathcal{R})), onehot(y)) \tag{35}$$

*where $CE(\cdot,\cdot)$ is the cross entropy loss, $onehot(y)$ is the one hot embedding of label $y$.*

**Proof** (Theorem 4). *Since $\arg\max_{m\in Y} f_m(x) \neq y$, $\arg\max_{m\in Y} g_m(x) = y$, there exist $i \in \{1,\ldots,c\}$:*

$$g_y(x) > f_y(x)$$
$$g_i(x) < f_i(x) \tag{36}$$

*Then we have:*

$$g_y(x) - g_i(x) \geq f_y(x) - f_i(x) \tag{37}$$

*Introducing Statement 4 of Lemma 1 and Lemma 5:*

$$g_i(Z^T(x,\mathcal{A};\mathcal{R})) - g_y(Z^T(x,\mathcal{A};\mathcal{R})) < Z_i^T(x,\mathcal{A};\mathcal{R}) - Z_y^T(x,\mathcal{A};\mathcal{R}). \tag{38}$$

*Moreover, since the prediction is not abstained.*

$$\mathbb{E}[g_y(x)] - f_y(x) > \mathbb{E}[g_j(x)] > f_j(x), \forall j \neq i, y \tag{39}$$

*This directly leads us to the result.*

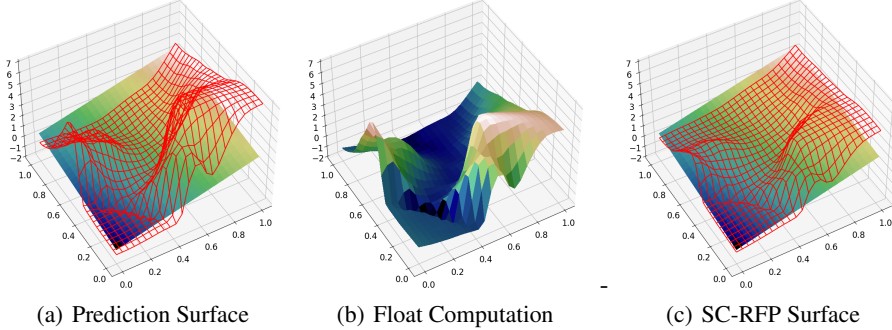

|  |  |  |
|---|---|---|
| (a) Prediction Surface | (b) Float Computation | (c) SC-RFP Surface |

Figure 6: The value of first element of prediction vector from VGG16 model trained on CIFAR10 given a 2D slice centered at a random data from test set. (a) The wireframe represents the prediction $f(x)$ while the surface is the sum of fixed path $Z^I(x, \mathcal{A}; \mathcal{R})$, respectively. (b) The sum of float path $Z^T(x, \mathcal{A}; \mathcal{R}) = f(x) - Z^I(x, \mathcal{A}; \mathcal{R})$. (c) The wireframe and surface are prediction of SC-RFP: $Z^I(x, \mathcal{A}; \mathcal{R}) + \eta Z^T(x, \mathcal{A}; \mathcal{R})$ with $\eta < 1$ and sum of fixed path same as (a).

Theorem 2 and Theorem 4 show that a smoothed classifier is not able to boost the performance of fixed path, while it is applied to reduce the sudden change of float path in a region. To be specific, if $x'$ can be corrected by $g$, it is resulted from the sudden change provided by the float path. A higher confidence score of $g(x')$ can be achieved by reducing the weight of float path during the computation path. On the other hand, if $x'$ is Lipschitz vulnerable, then regardless of the form of smoothed classifier, it cannot be fixed. This lead us to the theoretical basis of SC-RFP algorithm.

Figure 6 illustrates the SC-RFP by showing the prediction, fixed path and float path. We first present the prediction and the fixed path value in Figure 6(a). The float path value is then computed as $Z^T(x, \mathcal{A}; \mathcal{R}) = f(x) - Z^I(x, \mathcal{A}; \mathcal{R})$ in Figure 6(b). At last, by repressing the float value, we achieve a locally stable prediction from SC-RFP.

### C.2 VERIFIABLE RADIUS

At the end of section 4.2, we propose Theorem 3 to describe the upper and lower bound of SC-RFP as well as the certified radius. We present the proof of Theorem 3 below.

**Theorem 3.** *Let $N$ be a network defined as Section 3.1. Let $g$ be a smoothed classifier that samples noise from distribution $D_{noise}$ and $g'$ is the SC-RFP built on $g$. Assume that the direction of $\epsilon$ is uniformly distributed.*

$$\forall \|\eta_1\| = \|\eta_2\| = 1, P(\frac{\epsilon}{\|\epsilon\|} = \eta_1) = P(\frac{\epsilon}{\|\epsilon\|} = \eta_2), \epsilon \sim D \tag{40}$$

*If $\arg\max_{m \in Y} f_y(x) = y$, then*

$$\underline{p'_A} > \underline{p_A}, \overline{p'_B} < \overline{p_B}, \tag{41}$$

*where $\underline{p'_A}, \underline{p_A}$ are the lower bound of $g'_y(x)$ and $g_y(x)$, $\overline{p'_B}, \overline{p_B}$ are the upper bound of $g'_{m \neq y}(x)$, $g'_{m \neq y}(x)$. Moreover, $\arg\max_{m \in Y} g'(x) = y$ for all $\|\epsilon\| \leq R$,*

$$R = \frac{\sigma}{2}(\Phi^{-1}(\underline{p_A}) - \Phi^{-1}(\overline{p_B})) \tag{42}$$

**Proof.** *The proof of Theorem 3 can be divided into three steps. First, we show that given $\epsilon \sim D_{noise}$, the expectation of fixed paths for the any $\mathbb{E}(Z^X(x + \epsilon)) = Z^X(x)$. Consider $\zeta$ is a fixed path on a region R, then from Lemma 4*

$$E[g(\zeta(x))] = \zeta(x). \tag{43}$$

*Notice that, since the SC-RFP does not affect the sample of noise, then above equation holds for both $g$ and $g'$:*

$$E[g'(\zeta(x))] = \zeta(x). \tag{44}$$

*By applying randomized smoothing on statement 1 of Lemma 1, we have:*

$$E(g(x)) = \mathbb{E}(g(\mathcal{Z}^I(x, \mathcal{A}; \mathcal{R}))) + \mathbb{E}(g(\mathcal{Z}^T(x, \mathcal{A}; \mathcal{R}))),$$
$$E(g'(x)) = \mathbb{E}(g'(\mathcal{Z}^I(x, \mathcal{A}; \mathcal{R}))) + \mathbb{E}(g'(\mathcal{Z}^T(x, \mathcal{A}; \mathcal{R}))), \tag{45}$$

*where we use $g'(\mathcal{Z}^T(x, \mathcal{A}; \mathcal{R}))$ to denote applying smoothing algorithm g on deterministic function $\mathcal{Z}^I(x, \mathcal{A}; \mathcal{R})$. Since $\mathcal{Z}^I(x, \mathcal{A}; \mathcal{R})$ is the aggregation of all the fixed path, with Equation 43 and 44, we have :*

$$\mathbb{E}(g'(\mathcal{Z}^I(x, \mathcal{A}; \mathcal{R}))) = \mathbb{E}(g(\mathcal{Z}^I(x, \mathcal{A}; \mathcal{R}))) = \mathcal{Z}^I(x, \mathcal{A}; \mathcal{R}) \tag{46}$$

*This means that the difference between $E(g(x))$ and $E(g(x))$ is same with that of $g(\mathcal{Z}^T(x, \mathcal{A}; \mathcal{R}))$ and $g'(\mathcal{Z}^T(x, \mathcal{A}; \mathcal{R}))$.*

*Next, we discuss how the float path affect the prediction of $g(\mathcal{Z}^T(x, \mathcal{A}; \mathcal{R}))$ and $g'(\mathcal{Z}^T(x, \mathcal{A}; \mathcal{R}))$. Given $\epsilon \sim D_{noise}$, we denote $P_1$ as the event that $f_y(x + \epsilon) > f_y(x)$. Since we have excluded the fixed path, $P_1$ holds means that the float path boosts the probability of $f_y(x + \epsilon)$. However, as $\arg\max m \in Y f_y(x) = y$, repressing the float path does not alter the prediction of $f_y(x')$. On the other hand, we denote $P_2$ as the event that $f_y(x + \epsilon) < f_y(x)$. This means that the float path $\mathcal{Z}^T(x + \epsilon, \mathcal{A}; \mathcal{R}) < 0$ and negatively contributes to the prediction. Then repressing the float path means that can increase the probability of $y$.*

*In other words, for any $x + \epsilon$, repressing the float path between $x$ and $x'$ can increase the predicted $f_y(x + \epsilon)$. We assume that $P_1$ and $P_2$ happens with probability $p_1$ and $p_2$, then given a certain number of sampling,*

$$E(g'_y(x)) > E(g'_y(x)), Var(g'_y(x)) < Var(g_y(x))$$

*By applying Chebyshev Inequality, we have $\underline{p'_A} > \underline{p_A}$. Similarly, we also have $\overline{p'_B} < \overline{p_B}$. Therefore, we conclude the lower bound $p'_A$ is larger than that of $p_A$, and similar to $p'_B$.*

*Computing the certified radius of $x$ then is same with that of Cohen et al. (2019). Since both g and g' are random function that sample noise from same distribution, the Neyman-Pearson theorem holds for both g and g'. Therefore, the certified radius remains unchanged.*

## D    EXTRA TABLES AND FIGURES

This section present the complete experiment results. Table 4 shows the certified accuracy at different $l_2$ radius level.

Table 5 presents the experiment result of our model certified with different repression rate. Among the different repression rate, we find that when $\eta = 0.25$ SC-RFP provides the best performance on increasing the model robust accuracy for perturbation with large size, while the standard accuracy and are slightly damaged. This is also observed on previous works.

| $\sigma$ | Method | 0.25 | 0.5 | 0.75 | 1.0 | 1.25 | 1.5 | 2.0 | 2.5 | Clean |
|---|---|---|---|---|---|---|---|---|---|---|
| 0.25 | Cohen et al. (2019) | 66.0 | 50.2 | 44.2 | 0.0 | 0.0 | 0.0 | 0.0 | 0.0 | 75.3 |
|  | + SC-RFP | 69.4 | 58.3 | 52.7 | 0.0 | 0.0 | 0.0 | 0.0 | 0.0 | 77.2 |
| 0.50 | Cohen et al. (2019) | 53.4 | 44.7 | 34.0 | 25.4 | 19.5 | 19.5 | 0.0 | 0.0 | 60.3 |
|  | + SC-RFP | 57.3 | 50.5 | 42.4 | 34.9 | 29.5 | 29.5 | 0.0 | 0.0 | 62.7 |
| 1.00 | Cohen et al. (2019) | 31.2 | 27.1 | 23.2 | 19.3 | 15.3 | 11.9 | 6.9 | 4.5 | 34.7 |
|  | + SC-RFP | 36.0 | 32.6 | 29.4 | 25.9 | 22.2 | 18.4 | 12.9 | 9.5 | 39.1 |

Table 4: Certified robust accuracy for models with different methods on CIFAR10 dataset.

Figure 7 present the model trained with $\sigma = 0.05$ and $\sigma = 0.25$, but under different level of noise. We find the model trained with slightly noised sample can provide effective robustness.

| $\sigma$ | $l_2$ radius | 0.25 | 0.5 | 0.75 | 1.0 | 1.5 | 2.0 | Clean |
|---|---|---|---|---|---|---|---|---|
| 0.25 | Jeong & Shin (2020) | 59.8 | 49.8 | | | | - | |
| | Jeong et al. (2021) | 46.7 | 38.2 | | | | | - |
| | Cohen et al. (2019) | 57.9 | 46.4 | | | | | **66.1** |
| | + SC-RFP (ours) | 59.4 | 50.3 | | | | | 65.8 |
| | Salman et al. (2019) | 59.2 | 54.1 | | | | | 63.6 |
| | + SC-RFP (ours $\eta = 0.05$) | 59.6 | 54.5 | | | | | 63.8 |
| | + SC-RFP (ours $\eta = 0.10$) | 59.5 | 54.8 | | | | | 63.9 |
| | + SC-RFP (ours $\eta = 0.25$) | **59.8** | **55.4** | | | | | 63.6 |
| 0.50 | Jeong & Shin (2020) | **53.7** | **49.9** | 44.7 | 39.3 | | | - |
| | Jeong et al. (2021) | 39.1 | 34.9 | 30.3 | 26.8 | | | - |
| | Cohen et al. (2019) | 51.3 | 46.1 | 39.9 | 32.6 | | | **56.5** |
| | + SC-RFP (ours) | 51.1 | 47.3 | 41.5 | 35.6 | | | 55.9 |
| | Salman et al. (2019) | 52.6 | 48.7 | 44.0 | 39.6 | | | 56.0 |
| | + SC-RFP (ours $\eta = 0.05$) | 52.7 | 49.1 | 45.2 | 40.5 | | | 56.4 |
| | + SC-RFP (ours $\eta = 0.10$) | 52.9 | 49.4 | 45.4 | 40.8 | | | 56.2 |
| | + SC-RFP (ours $\eta = 0.25$) | 52.8 | 49.6 | **46.0** | **42.0** | | | 55.2 |
| 1.00 | Jeong & Shin (2020) | 40.0 | **38.5** | 35.4 | 32.6 | 28.1 | 22.6 | - |
| | Jeong et al. (2021) | 29.7 | 26.2 | 23.0 | 20.6 | 15.7 | 12.1 | - |
| | Cohen et al. (2019) | 39.1 | 36.1 | 32.5 | 29.1 | 22.8 | 15.6 | **42.4** |
| | + SC-RFP (ours) | 39.9 | 36.5 | 33.9 | 31.5 | 25.6 | 19.9 | 42.9 |
| | Salman et al. (2019) | 39.5 | 37.1 | 34.7 | 31.8 | 26.2 | 20.1 | 42.0 |
| | + SC-RFP (ours $\eta = 0.05$) | **40.2** | 37.5 | 35.2 | 33.3 | 28.0 | 21.8 | 42.2 |
| | + SC-RFP (ours $\eta = 0.10$) | 39.9 | 38.1 | 35.7 | 33.4 | 28.9 | 23. | 41.9 |
| | + SC-RFP (ours $\eta = 0.25$) | 39.7 | 37.8 | **35.9** | **34.2** | **29.6** | **25.2** | 41.6 |

Table 5: Certified robust accuracy for models with different methods on ImageNet dataset

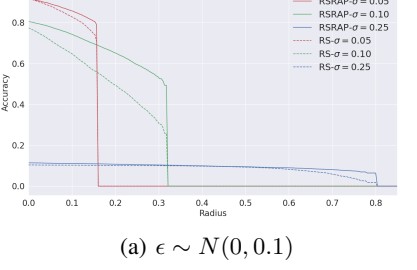

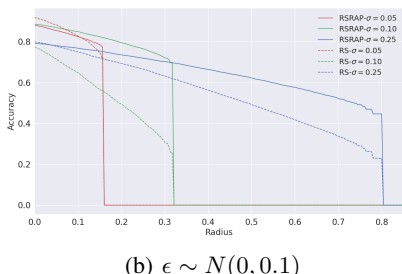

(a) $\epsilon \sim N(0, 0.1)$            (b) $\epsilon \sim N(0, 0.1)$

Figure 7: Certified Accuracy of base methods and SCRFP ($\eta = 0.1$) with different level of noise on CIFAR10. The solid and dashed lines represent benchmark and SC-RFP. The models are tested under different level of noise.

