# OpenReview forum: "On Explaining Neural Network Robustness with Activation Path"
_ICLR.cc/2023/Conference — ICLR 2023 poster_

### Official Review · Reviewer_2h1E · 2022-10-17

**Confidence:** 3
**Correctness:** 3
**Technical Novelty And Significance:** 3
**Empirical Novelty And Significance:** 3
**Recommendation:** 6

**Clarity, Quality, Novelty And Reproducibility:**

The idea behind SC-RFP is simple and novel (to the best of my knowledge). However, the presentation is quite confusing, full of typos, and assumes deep familiarity with concepts in the randomized smoothing literature. The notation is particularly heavy and, in my opinion, convoluted, given that some of the employed concepts are well-known (like activation patterns). The paper introduces non-standard concepts without proper explanations: for instance, "bent-hyperplanes", which likely comes from Hanin et al 2019b.

The related work is relatively comprehensive. However, it would be nice if the authors could acknowledge work on bridging the gap between adversarial training and verified training ([ReLU stability](https://arxiv.org/abs/1809.03008), [COLT](https://openreview.net/forum?id=SJxSDxrKDr), [IBP-R](https://arxiv.org/abs/2206.14772), as these works try to address some of the scalability issues of worst-case verified training.

The authors provide code in the supplementary material, ensuring reproducibility (though I have not checked the code itself).

**Strength And Weaknesses:**

The authors divide neurons of piecewise-linear networks into "fixed" and "float" neurons, depending on whether their activation phase changes within the perturbation radius, and point out that randomized smoothing can only affect the contribution of float neurons to adversarial vulnerability. Starting from this rather intuitive observation, the authors propose to keep track of float neurons, and to reduce their effect, hence further smoothing the resulting classifier.

The empirical performance of the method (at seemingly no additional cost) is definitely appealing. The method only relies on simple intuition (a strength), but does not (as far as I understand) provide any theoretical result backing the empirical performance.
The randomized smoothing literature (e.g., Cohen et al 2019) provides a tight upper bound on the robustness of the smoothed classifier. Does SC-RFP alter this upper bound?
In Cohen et al 2019 the reported certified accuracy is the "approximate certified test set accuracy", which is a high-probability lower bound on the true certified accuracy. Is this the case for the provided results? Could the authors provide additional information on which statements from Cohen et al 2019 hold for SC-RFP, along with a short proof? (I believe porting some of the results will be quite immediate)
In general, the poor presentation of the paper makes it hard to adequately assess the authors' claims.



**Summary Of The Paper:**

The paper presents SC-RFP, a novel algorithm to enhance the empirical performance (certified accuracy) of classifiers based on randomized smoothing. SC-RFP operates by reducing the contribution of neurons that change activation phase across perturbations, and is demonstrated to positively affect performance.

**Summary Of The Review:**

While the algorithmic idea behind SC-RFP is novel and empirically effective, I do not believe the paper is publishable in the present form due to the quality of its presentation. However, I am willing to increase my score if the authors addressed the major weaknesses of the presentation.

---

> ### Author Response · Authors · 2022-11-18
> **Thanks for your comments (1)**
>
> Dear Reviewer 2h1E:
>
> We are thankful for your confirmation of the novelty, and empirical performance of this paper and, more importantly, advice on providing additional theoretical results to back the empirical performance of our algorithm. As for your concerns, we would like to address them as follow:
>
> 1. Strength And Weaknesses:
> Upper Bound of Certified Accuracy: Yes, the upper bound from Cohen et al (2019) holds for our SC-RFP. We would like to thank you for pointing out the presentation issue that prevents readers from fully capturing the proof of this work. We revised the entire Section 4.3 and move some of the Theorems to the appendix. We also build a connection between SC-RFP and Theorem 1 in (Cohen el al) to show that the certified radius also holds for our algorithm. This is presented as Theorem 3 in the revised version.
>
> The idea in our draft is:
> In Section 4.3, we discuss two kinds of adversarial examples: The Lipschitz vulnerable example and the float neuron vulnerable example.
> The first type is caused by the locally large Lipschitz constant. For example, the local Lipschitz constant of the model without any defenses can reach hundreds, this means that with small perturbation $\epsilon$, the $x + \epsilon$ can mislead the classifier. Theorem 1(old) shows for such a data $x$, there always exists an adversarial example $x' \in B(x, r)$ that can mislead a randomized classifier. In another word, a randomized classifier cannot correct this kind of error.
>
> The second type of error, on the other hand, can be cured by the randomized classifier. We propose Theorem 2(in old version) to show that, under the assumption that $x' \in B(x, r)$ is misclassified by base classifier $f$ while corrected by smoothed classifier $g$, SC-RFP can provide better results on such an example $x'$.  Based on Theorem 1(old) and 2(old), Theorem 3 (old)concludes that SF-RFP can provide a better certifiable accuracy under the same criteria.
>
> Due to the page limitation, we delete some parts of our discussion in the original version which causes you confuse about our statement. To address this, we (1) modified the discussion in Section 4.3 to better describe our motivation (2) address the major concern you mentioned with (new)Theorem 3, which ensures the upper bound from Cohen et al (2019) holds for SF-RFP, and explain that at the end of section 4.3 paragraph. We believe this can improve the readability of this work and (3) we move Theorem 2 and 3 to the appendix.

---

> > ### Author Response · Authors · 2022-11-18
> > **Thank you for your comments (2)**
> >
> > 2. Clarity, Quality, Novelty, And Reproducibility:
> >
> > Still, we are grateful for your suggestion on the presenting issue. We realize that we need an additional paragraph in the main paragraph to clarify our motivation for definitions and theorems. The detailed modifications are:
> >
> > 2.1 Randomized Classifier: We realized that for readers not familiar with randomized smoothing, it can be a bit confusing, therefore, we (1) added an additional paragraph in Section 3.2 (Randomized Certifiable Classifier) to discuss the details of smoothed classifier; (2) modified our description so that it in line with previous works, (3) include more details for state-of-art randomized smoothing methods.
> >
> > 2.2 Activation Pattern: We believe your concern about the heavy notation is fair and useful in improving the presentation of this work. To address the issue, we (1) add additional discussion following the definition and state the necessity of those notations, (2) delete some of the confusing terms while adding more explanation about the remaining.
> > There are 3 major differences between our definition and previous works (for instance, Hanin 2018):
> > (1) We assign each neuron an index, while previous works use "z is a neuron in $N$". The reason is that we have to specify the activation pattern for each neuron when decomposing the computational graph in Lemma 1.
> > (2) We use a introduce additional notation $U=\{U_1, U_2, ..., U_n\}$ for the regions of activation function. There are two reasons for that: First, we are currently working on generalizing the investigation to continuous non-linear functions. For instance, the GeLU activation can be split into 3 regions by the set $\{\mbox{minimum point}, 0\}$. The first region is semi-linear decreasing, the second is semi-linear increasing with a lower slope, and the last one is semi-linear increasing with a larger slope. By relaxing an upper and lower bound on each region, we can also provide a relaxed bound on those networks with those functions. Second, similar to 1, it is necessary to specify the slope of each region when decomposing the computational graph, which we worried about using $\{-1, 1\}$ (as in Hanin 2018) to indicate the activation region might cause more confusion.
> > (3) We introduce an operator $\hat{a}_{ij}(x): x \rightarrow \{0, 1, ..., q\}$ as a mapping from an input to the pattern of neuron (i,j), while Hanin(2018) defines activation region based on the pattern, but did not define the inverse operator. As their objective is to study the expressive ability of neural networks and mostly focus on the number of regions as well as the properties of regions, such a definition is not necessary for them.
> >
> > However, in our work, the definition of float / fixed neuron is space dependent. In different subspaces $R_1, R_2 \subset R^{n_0}$ of the input space, the fixed and float neurons can be different. Informally, definition 4 means "a neuron has same activation pattern for every $x\in R \subset R^{n_0}$ is a fixed neuron in $R$, otherwise, it is a float neuron". When it comes to the formal definition, it is necessary to declare what is "activation pattern of $x$".
> >
> > Apart from that, there is a possible flaw when defining the "activation pattern of $x$", when the pre-activation of neuron (i,j):$z_{ij}$ equals the breakpoint (for ReLU, $z_{ij}=0$). In order to present rigorous statements, we include this case in the definition 1 Equation 6.
> >
> > This work is greatly inspired by previous works in explaining neural networks with activation patterns (such as Hanin 2018), and we tried to illustrate our concepts on their framework. However, we found that there are several issues, as mentioned above, that can cause more severe misunderstandings. Therefore, we introduced additional notations.
> >
> > 2.3 bent-hyperplane: Yes, it is from Hanin (2018, 2019b). We use this concept to describe how input spaces are partitioned into activation regions. The objective is to provide a geometric intuition and prelude for Lemma 2. We would like to thank you for your comments which make us realize that this can result in a clarity issue. To address this, we added a paragraph to discuss it in Section 3.3.
> >
> >
> > 2.4 Related works: We would like to thank you for filling up the missed related works. We added them to the revised version (Section 2) and states their contribution to the literature.

---

> > > ### Comment · Reviewer_2h1E · 2022-11-18
> > > **Thanks for improving the quality of the presentation**
> > >
> > > I thank the authors for improving the quality of the presentation and their detailed response. I increased my score to 6.

---

> > > > ### Author Response · Authors · 2022-11-19
> > > > **Thank you for your approval of the revised version.**
> > > >
> > > > Dear Reviewer 2h1E:
> > > > We would like to thank you for your approval of the revised version. The advice you proposed is really helpful for us to address the flaws of our draft.

---

### Official Review · Reviewer_EMXC · 2022-10-21

**Confidence:** 2
**Correctness:** 4
**Technical Novelty And Significance:** 3
**Empirical Novelty And Significance:** 2
**Recommendation:** 6

**Clarity, Quality, Novelty And Reproducibility:**

The paper seems reproducible.

I have not checked through every detail of the proof in the appendix, but the lemma and theorem statements seems reasonable to me.

**Strength And Weaknesses:**

Strength:
- Claims are well supported and the design of SC-RFP is reasonable.
- SC-RFP provides decent improvement over some prior work.

Weakness:
- One major concern is the lack of comparison with related work [1]. I believe this work also decompose the ReLU network in to regions formed by linear hyperplanes (although they are not based on smoothing).
- What is the shift parameter of \phi_i?

[1] Jordan, Matt, Justin Lewis, and Alexandros G. Dimakis. "Provable certificates for adversarial examples: Fitting a ball in the union of polytopes." Advances in neural information processing systems 32 (2019).

Minor:
- For the definition of U_i in page 3, should s_i be \gamma_i instead?
- There are many notations, having some figures to make the definition and lemmas more intuitive would be great.


**Summary Of The Paper:**

This paper begins by defining the activation path/neuron and categorize them into fixed and float path/neuron. Then they show that smoothed classifiers cannot boost the performance of fixed paths. Based on this observation, they then proposed their algorithm -- SC-RFP. SC-RFP down weights the float paths, in other words, they want the smooth out the float paths. Experimentally, they report the proportion of fixed neurons and concluded that majority of the neurons are not affected by SC-RFP. They also show that their method (SC-RFP) can achieve a better certified robustness on ImageNet comparing with many recent works.

**Summary Of The Review:**

This is a solid paper. The reasoning makes sense, and the proposed algorithm has decent improvement.

---

> ### Author Response · Authors · 2022-11-18
> **Thank you for your comment**
>
> Dear Reviewer EMXC:
>
> We would like to thank you for your comments and confirmation of the solidity of this paper. We hereby respond to the mentioned weakness and flaws:
>
> Weakness.1: Thank you for your suggestion on the missing literature. We have included the mentioned paper in the literature review section. This work introduces an algorithm named GeoCert which finds and computes the $l_p$ bound of the network with a piecewise linear activation function. As the objective of [1] is to find sound robustness bound for the network while ours focus on improving randomized smoothing, we cannot compare those algorithms. However, this work shed light on investigating the neural network robustness on the basis of activation regions.
>
>
> Weakness.2: Thanks for pointing out the clarity issue of Lemma 1. We rephrased the statement should as "$W'^{(i)}$ and $\beta^{(i)}$ are the matrix and bias of linear transformation in layer $i$". The $W^{(i)}$ and $\beta^{(i)}$ are the trivial weight matrix and bias for fully-connected layers. For complex cases, such as convolutional layers or layers with batch normalization, as the transformation is still linear, the layer can also be represented by a weight matrix $W'^{(i)}$ and bias $\beta^{(i)}$.
>
> Weakness.Minor.1: Yes, it should be. We changed some of the notation after finishing the first draft and left some of the typos there. We would like to thank you for pointing that out. We revised it in the modified version and double-checked other notations.
>
> Weakness.Minor.2: Thanks for your suggestions, we added a demonstration in Appendix A with additional discussion to illustrate the concept. The toy model describes how the input space is partitioned into pieces and the relationship between different regions.

---

### Official Review · Reviewer_Aniy · 2022-10-23

**Confidence:** 3
**Correctness:** 3
**Technical Novelty And Significance:** 3
**Empirical Novelty And Significance:** 3
**Recommendation:** 6

**Clarity, Quality, Novelty And Reproducibility:**

I found this paper a bit hard to follow given the notation issues mentioned in the weakness section above. I humbly suggest the authors double-check the notations and simplify some notations for better readability. I also suggest the authors have a diagram of a toy example to better explain the concepts introduced.

**Strength And Weaknesses:**

Strength:
Recently researchers started to look at the activation space of neural nets for adversarial robustness. This paper is also from this perspective but combines it with randomized smoothing.

Weakness:
1. I found some notation issues in the paper. For example:
1.1 in equation (4), the expectation $E[g_m(x')=y]$ is unclear to me. Should there be an indicator function, as $1_{g_m(x')=y}$ in the expectation? Also, I think it should be $\forall x'\in B $ instead of $\forall x\in B $.
1.2 In Definition 1, the meaning of $(i, j)\in \mathcal{I}$ also seems unclear to me (does that mean "index"?). I think it would be better to define it in the statement.
1.3 In Equation (8), the authors used $\cap$ (intersection) instead of $\cup$ (union), which contradicts the sentence above it. I assume the sentence above Eq (8) should be "intersection".
1.4 It seems to me that the authors used $\mathcal{R}$ and $\mathbb{R}$ interchangeably for the activation region.
2. In the proof of Lemma 1 the authors use batch normalization, but the conditions of Lemma 1 do not include it. I am wondering why batch normalization is needed in the proof.
3. Statement 4 of Lemma 2 is unclear to me. Proof of it in the appendix is also not so convincing. Do the authors assume the activation is always ReLU in this Lemma, or it can be any function such as sigmoid? If it is not linear, I don't think we can replace a finite difference with a simple Jacobian, which is claimed in the proof. Also, even if its a fixed path, the output can also change because the activation is not constant. So the statement that $Z^I(x',A,R)=Z^I(x',A',R)$ is not clear to me. Did I miss anything here?
4. I think there is an important reference missed: Zhang et al "A Branch and Bound Framework for Stronger Adversarial Attacks of ReLU Networks", ICML 2022. This work also studied the adversarial examples in the activation space of ReLU networks.

**Summary Of The Paper:**

This paper proposed a framework to decompose a neural network based on its neurons' activation status. Then the authors tried to explain adversarial examples and randomized smoothing based on this framework, and proposed a method to repress the float path to achieve better certified robustness.

**Summary Of The Review:**

In general, I think the author's high-level intuition is interesting and this is a good research perspective. But I found some flaws in the author's claims. So I encourage the authors to clarify them in a revised version.

---

> ### Author Response · Authors · 2022-11-18
> **Thanks for your comments**
>
>
> Dear Reviewer Aniy:
>
> We are grateful for your thorough review and for pointing out the potential issues in the draft. We are glad that you find our research an interesting topic, and would like to address your concerns in the following:
>
> Weakness.1.1: Thank you for pointing out the notation conflicts between Equations (3) and (4). In Section 2.1 we declare the function $f$ to be a mapping from $R^{n_0} \rightarrow R^{c}$, therefore we use $argmax_m f_m(x)$ for the classification. However, in Equation (3) we were thinking of following the notation of Cohen et al(2019) to denote the output of the randomized classifier $g$, where the output of $g$ is the predicted label. This results in a clarity issue in this work.
> As for "$x \in B$", it should be $x'$ like you suggested.
> In the latest version, we revised our statements so that it is in line with statements of previous works.
>
> Weakness.1.2: Yes, (i,j) are referred as to the $j$-th neuron in layer $i$, which we specified in the notation Section (at end of Page 2 and beginning of Page 3): "Let $n_i$ be the output size of layer $i$, we use a tuple pair $(i,j)$ to denote the $j$-th component of layer $i$ and denote $\mathcal{I} = \cup_{i=1}^{d-1}\{(i,j) | j \in \{1, \dots, n_{i}\}\}$ as the collection of the indexes of intermediate layers." We realized that there are potential clarity issues, and revised the statements in all the Lemmas to refer the network structure to the notation section.
>
> Weakness.1.3: Yes, it should be an intersection, we fixed it in the revised version.
>
> Weakness.1.4: We realized that there are some conflicts between the pre-defined macro command in our draft of the ICLR template, which resulted in the typo you mentioned. We have fixed it and double-checked other typos.
>
> Weakness.2: Actually, the proof of Lemma 1 holds regardless of the existence of the batch normalization layer. As stated in the notation section, we declare that there exist optional batch-normalization layers in order to meet the empirical. As you pointed out, we realized that there might be an unclarity in our definition. We add a notation section in the revised version and revised all the lemmas, theorems, and definitions so that they are all in line with the statements in the notation section.
>
> Weakness.3: Yes, our discussion is based on a neural network with a piecewise linear activation function. We state that in the Introduction section: "As the theoretical basis is built on piecewise linear functions, in the following of this work, we assume network $N$ has piecewise linear activation without specification.". However, we do notice that this is not very obvious. Therefore, we revised the notation part in Section 2 to address the clarity issue. We would like to thank you for bringing that to our attention.
>
> We are glad to see that you mentioned the non-linear case. In the future, we are considering expanding our discussion to the non-linear case. With a certain upper bound and lower bound for the activation region of the non-linear activation function. It is possible to generate an upper and lower boundary for the non-linear activation regions.
>
> Last, we would like to answer your question regarding the proof of Lemma 2. $\mathcal{Z}^{I}(x', \mathcal{A}; \mathcal{R})$ and $\mathcal{Z}^{I}(x', \mathcal{A}'; \mathcal{R})$ are the fixed path in the set $\{x. x'\}$. As Definition 3 and 4 suggest, all the neuron along the fixed path has identical activation pattern, which ensures that the activation pattern of each neuron along a single path is identical for $x$ and $x'$. (In Figure 2, we show that even in a network trained with $\sigma=0.05$, there are around 90\% of neurons are fixed between $x$ and $x'$).
>
> Combing with Lemma 1, if all the neuron on a path has identical activation pattern, then the function in that subspace (in this case, $\{x, x'\}$) is linear. Since $\mathcal{Z}^{I}(x', \mathcal{A}; \mathcal{R})$ and $\mathcal{Z}^{I}(x', \mathcal{A}'; \mathcal{R})$ are the aggregation of all the fixed path between $\{x, x'\}$, we have $\mathcal{R}) = \mathcal{Z}^{I}(x', \mathcal{A}'; \mathcal{R})$. To make this clear, we add an additional lemma in the proof section to increase the readability.
>
> Weakness.4: We would like to thank you for the reference. This work introduces an algorithm that systematically searches the adversarial example based on the activation space of the ReLU network. We included it in the discussion of state-of-art techniques of adversarial examples in the Related Works Section.
>
> Clarity, Quality, Novelty, And Reproducibility: To improve the readability of this work, we added an illustration of the fixed(float) neuron(path) in the appendix section (due to the page limit) as you suggested. Following the Definitions, we refer this part to the reader so that it can be easier for the reader to capture the intuition of this work.

---

> > ### Comment · Reviewer_Aniy · 2022-11-20
> > **Revised score**
> >
> > Thanks for the revisions and they addressed my concerns. I turned my score to positive and would ask AC to decide. I think it's better to put figure 5 into the main text, if possible.

---

> > > ### Author Response · Authors · 2022-11-21
> > > **Thank you for your approval of our revision**
> > >
> > > Dear Reviewer Aniy:
> > > We would like to thank you for your approval of our revision. We will change the order of paragraphs (move some of the discussion into the appendix) and try to fit the figure in the main text in the next revision.

---

### Official Review · Reviewer_UkvR · 2022-10-27

**Confidence:** 2
**Correctness:** 3
**Technical Novelty And Significance:** 4
**Empirical Novelty And Significance:** 3
**Recommendation:** 6

**Clarity, Quality, Novelty And Reproducibility:**

Clarity: The illustration of the definition and lemma is not very clear. It lacks an overview of how the theorems and lemmas are connected. For example, I am not sure how lemma 1 contributes to deification 3 and 4, which makes the paper not easy to follow.

Quality: There are also some typos and wrong definition in the paper. Such as \mathbb{R} is also in the text however \mathcal{R} is used in the equation.

Novelty: The proposed idea is novel and interesting and could be impactful to understand the model's robustness.

Reproducibility: Since the hyperparameters and code are provided, the paper results should be reproducible.

**Strength And Weaknesses:**

Pros:
1. The proposed idea is quite novel and could be impactful to understand the model's robustness.
2. The proposed method is backed with comprehensive theoretical analysis.
3. The proposed method shows a clear improvement on the clean accuracy.


Cons:
1. It is not clear whether the proposed assumption that there only exists fixed and float path is true. Or in other words, it is not trivial to set a threshold to determine which neurons would be regarded as float neurons or fixed neurons. The BinomPValue is used however it is not sure it matches the definition in the theoretical analysis.

**Summary Of The Paper:**

The paper proposes to divide computational graph of the neural network into the fixed path and float path and then shows the improvement of robust accuracy is mainly gained from correcting the vulnerability floating neurons. Specifically, the neural network graph is first divided into several activation regions and thus the output of layer i could be decomposed into the combination on those regions. Then based on whether the prediction would change or not, it defines the float and fixed neurons. The float path then could defined if there exists any neurons in the paths. According to the definition, the margin change in the output could be found not related with the fixed path. Therefore, the paper proposes SCRFP that empirically count those neurons that the change of prediction is in a certain region by adding Gaussian noise. The experiment shows the proposed method achieve a better clean accuracy with the same robust radius.

**Summary Of The Review:**

The paper proposes an interesting perspective to understand different neurons' roles in the random smoothing procedure and then proposes a way to increase the performance of random smoothing. The proposed method is novel and could be impactful. However, the paper lacks clarity on how the lemma and definition and make the paper not easy to follow.

---

> ### Author Response · Authors · 2022-11-18
> **Thanks for your comments**
>
> Dear Reviewer UkvR:
> Thank you for your comments on our work and your confirmation of our novelty as well as contributions. As for concerns, we would like to address them as follow:
> Cons.1: Given a network $N$, The set of float neurons and fixed neurons are defined on a subspace $R \in R^{n_0}$ of the input domain. If a neuron with index $(i,j)$ has the same activation pattern for every $x \in R$, then neuron $(i,j)$ is a fixed neuron. Otherwise, it is a float neuron. We would like to thank you for pointing out the possible unclarity and, to fix this, we revised our discussion on page 4 and page 5.
> We also added additional discussion in Appendix A with an illustration to show that a neuron is either fixed or float according to the definition.
>
> As for the threshold you mentioned, we would like to explain it as follows: given a piecewise linear activation function $\pi$, the threshold is determined naturally by the breakpoints of $\pi$. For instance, $\{0\}$ splits the ReLU activation function into activated and deactivated regions, while $\{0, 6\}$ splits the ReLU6 activation function into 2 deactivated regions (with slope 0) and 1 activated region (with slope 1).
>
> The theoretical analysis is provided in the revised version with proofs in Appendix. We show that the upper and lower bound from previous work (Cohen et al (2019)) also holds for our method. With this, the BinomPValue can be applied as that in previous works. We realized that in the original version of this work, there exists a representation issue. Therefore, we revised the entire section 4 and put the key result at the end of that section (theorem 3).
>
> Clarity.1: We would like to thank you for your suggestion. To address this issue, (1) we move Definition 3 and 4 to the beginning of Section 4.1(new section) and discuss their connection with Definition 2.  (2) Add an additional paragraph to discuss the motivation of Definition 3 and 4. We put Lemma 1 (in the old version) to Section 4.2 and discuss why the float path is useful in decomposing the network. as well as how we followed your suggestion and revised the discussion between Lemma 1 and Definitions 3 and 4.
>
> Quality.1: We realized that there are some conflicts between the pre-defined macro in our draft of the ICLR template, which resulted in the typo you mentioned. We have fixed it and double-checked other typos.
>
> Novelty.1: We would like to thank you for your confirmation of our novelty.

---

### Author Response · Authors · 2022-11-21
**Thank all the reviewer for your comments on this work.**

We would like to thank all the reviewers for their valuable advice. Based on this, we revised and improved the quality of this work. The changes include:

(1) Section 2, additional discussion on related works.
(2) Section 3.1, revising the definitions.
(3) Section 3.2, improving the discussion of the smoothed classifier with more details.
(4) Section 3.3, adding extra paragraphs to discuss the motivation of proposed definitions.
(5) Section 4.1, changing the order of some definitions and theorems.
(6) Section 4.2, additional discussion on the motivation of lemma and theorem.
(7) Section 4.3, theoretical analysis of the upper/lower bound of the proposed method, which is in line with previous works.
(8) Appendix A, an illustration of the geometrical intuition of proposed concepts.
(9) Appendix A-C, additional discussion on the theorems and lemmas.

---

### Decision · Program_Chairs · 2023-01-20

**Decision:**

Accept: poster

**Justification For Why Not Higher Score:**

There are several rooms for the improvements, as pointed out in the meta review.

**Justification For Why Not Lower Score:**

The paper makes a solid contribution to the randomized smoothing algorithm.

**Metareview: Summary, Strengths And Weaknesses:**

The paper first demonstrates an interesting finding to explain why randomized smoothing works --- the paper partition neurons into "float neurons" and "fixed neurons" by whether the activation pattern of the neuron changes within an epsilon ball, and shows that randomized smoothing can lead to a robust classifier mainly based on correcting float neurons. Based on this observation, the authors proposed a simple algorithm to improve the certified robustness of the randomized smoothing algorithm by repressing the float neurons.

All the reviewers and the AC think this is a strong paper and should be published. The finding of "randomized smoothing can't correct Lipschitz vulnerable data but can correct float neuron vulnerable data" is interesting. Further, the proposed algorithm seems to be a universal technique that can be easily applied to any existing randomized smoothing methods. Therefore, we recommend accepting this paper.

During the discussions, some of the reviewers still think that the clarity of the paper can be improved. More specifically, the definition of float/fixed neurons/paths may be over-complicated and we'd like to encourage the authors to simplify the notation in the final version, if possible. Further, the AC has the following suggestions after reading the paper and discussing with reviewers:

- Theorem 2: At the first glance, it is hard to understand why Theorem 2 implies "the smoothed classifier fails to correct Lipschitz vulnerable data". Although it becomes clearer after careful thinking, please add more explanations here since this is one of the main contribution of the work. Also, please use the same font for Z^T as defined in Theorem 1.

- Connection between observation and theory: The connection between the observation (Theorem 2) and the proposed algorithm is not very clear. In fact, the AC still has some questions even after reading the response from the author. In particular, the authors mentioned that

"As we discussed, the smoothed classifier aims to reduce the effect of that sudden change from the float path, this implies that repressing the float path can further improve the performance of the smoothed classifier."

I understand the first sentence is implied by Theorem 2, but it's still not clear why this implies repressing the float path can *further* improve the performance. It's possible that the randomized smoothing can already correct such sudden changes and thus repressing the float paths has little effect on the final performance. We encourage the authors to add more discussions about the connection between Theorem 2 and the proposed algorithm.





**Note From Pc:**

if the above contains the word "oral" or "spotlight" please see: "oral" presentation means -> notable-top-5% and "spotlight" means -> notable-top-25%. As stated in our emails, we are disassociating presentation type from AC recommendations